# Species Differences in the Biotransformation of Aflatoxin B1: Primary Determinants of Relative Carcinogenic Potency in Different Animal Species

**DOI:** 10.3390/toxins17010030

**Published:** 2025-01-09

**Authors:** David L. Eaton, David E. Williams, Roger A. Coulombe

**Affiliations:** 1Department of Environmental and Occupational Health Sciences, School of Public Health, University of Washington, Seattle, WA 98195, USA; 2Environmental and Molecular Toxicology, College of Agricultural Sciences, Oregon State University, Corvalis, OR 97331, USA; david.williams@oregonstate.edu; 3Graduate Toxicology Program, Department of Veterinary Sciences, Utah State University, Logan, UT 84322, USA; roger@usu.edu

**Keywords:** aflatoxin, biotransformation, species differences, cytochrome P450, glutathione S-transferase, biomarkers

## Abstract

It has been known since the early days of the discovery of aflatoxin B1 (AFB1) that there were large species differences in susceptibility to AFB1. It was also evident early on that AFB1 itself was not toxic but required bioactivation to a reactive form. Over the past 60 years there have been thousands of studies to delineate the role of ~10 specific biotransformation pathways of AFB1, both phase I (oxidation, reduction) and phase II (hydrolysis, conjugation, secondary oxidations, and reductions of phase I metabolites). This review provides a historical context and substantive analysis of each of these pathways as contributors to species differences in AFB1 hepatoxicity and carcinogenicity. Since the discovery of AFB1 as the toxic contaminant in groundnut meal that led to Turkey X diseases in 1960, there have been over 15,000 publications related to aflatoxins, of which nearly 8000 have addressed the significance of biotransformation (metabolism, in the older literature) of AFB1. While it is impossible to give justice to all of these studies, this review provides a historical perspective on the major discoveries related to species differences in the biotransformation of AFB1 and sets the stage for discussion of other papers in this Special Issue of the important role that AFB1 metabolites have played as biomarkers of exposure and effect in thousands of human studies on the toxic effects of aflatoxins. Dr. John Groopman has played a leading role in every step of the way—from initial laboratory studies on specific AFB1 metabolites to the application of molecular biomarkers in epidemiological studies associating dietary AFB1 exposure with liver cancer, and the design and conduct of chemoprevention clinical trials to reduce cancer risk from unavoidable aflatoxin exposures by alteration of specific AFB1 biotransformation pathways. This article is written in honor of Dr. Groopman’s many contributions in this area.

## 1. Introduction

The initial discoveries of aflatoxin as a potent hepatotoxin among turkeys, ducks, and certain strains of trout in the early 1960s provided the first insights into the potential long-term risk of cancer in humans [1,2,3,4]. However, tumors were not evident in the turkeys, perhaps because the potent hepatotoxic effects caused mortality well before tumors had a chance to develop. However, at approximately the same time, a strain of rainbow trout was identified as susceptible to the toxicities of *A. flavus*-contaminated feed, including the occurrence of liver tumors. Even just 5 days of exposure to 20 ppb aflatoxin B1 (AFB1) in the diet of young rainbow trout resulted in a 12% incidence of liver tumors 12 months later [2]. Soon after, rainbow trout were identified as perhaps being the most sensitive animal to the carcinogenic effects of AFB1, wherein trout fed a diet containing only 8 parts per billion of AFB1 developed liver tumors at a 70% incidence 12 months later [1]. The potent hepatocarcinogenic effects of purified AFB1 in mammals was demonstrated in 1967 by Wogan and Newberne [3]. A few years later they completed a more in-depth dose–response lifetime assessment of the hepatocarcinogenicity of aflatoxin in male Fisher rats, further highlighting the potent, dose-related carcinogenic effects of AFB1 in mammals [4]. However, Wogan’s group also found that a standard strain of laboratory mice (Swiss mice) seemed completely resistant to the carcinogenic effects of AFB1, with doses as high as 150,000 ppb producing no tumors [5].

Although there are likely numerous fundamental biochemical differences that contribute to such remarkable species differences in susceptibility to the carcinogenic effects of aflatoxin, it is now well recognized that such differences are predominantly related to specific biotransformation pathways. The biotransformation of AFB1 is somewhat complex, involving multiple steps of oxidation, reduction, conjugation, and macromolecular adduct formation. Although the metabolic pathways that AFB1 undergoes are qualitatively similar in most, if not all, vertebrate species, there are important quantitative differences in the expression and catalytic activity of the enzymes responsible for AFB1 biotransformation that impart large differences in the relative species sensitivity to the carcinogenic—as well as other toxic—effects of AFB1. While aflatoxin metabolism has been reviewed extensively over the past decade (see the following references as examples: [6,7,8,9,10,11,12,13,14,15,16,17]), this review provides a detailed historical context to the development of our current understanding of the important role that subtle differences in both the expression and catalytic activity of a variety of biotransformation enzymes play in determining relative species susceptibility to the carcinogenic effects of AFB1.

## 2. Phase I Biotransformation of AFB1: Oxidation Reactions

There are numerous different oxidative pathways for AFB1, nearly all of which are mediated via one or more cytochromes P450 (CYPs; Figure 1).

### 2.1. Step #1 Oxidation of AFB1 to Aflatoxin-8,9-Epoxide

There is little question that the initial oxidation of AFB1 to the unstable aflatoxin B1-8,9-epoxide (AFBO) represents the single most important step involved in most, if not all, toxic effects, including carcinogenesis. There are two stereoisomers of AFBO, designated ‘exo-’ and ‘endo-’ AFBO, where the epoxide structure is either above or below the plane of the rest of molecule. Only the exo- isomer reacts with DNA [18]. The importance of the double bond in the cyclopentanone ring is illustrated by the much lower toxicity and carcinogenicity of aflatoxin B2 (AFB2) [19], which is chemically identical to AFB1 except that this double bond is absent.

The oxidation of AFB1 to AFBO is mediated almost exclusively by the cytochrome P450 (CYPs) family of enzymes. There is a large body of in vitro and in vivo data on liver microsomal biotransformation of AFB1 in a variety of species. Ramsdell and Eaton [16] compared the initial rates of oxidation of AFB1 in human, rat, mouse, and Macaca liver microsomes under identical experimental conditions, using a high AFB1 concentration of 124 µM (Figure 2). Formation of AFBO, measured by trapping the epoxide with BHA-induced mouse liver cytosol, occurred in the rat at a rate of 0.6 nmol of product/mg of microsomal protein/min, which was about 3 times higher than human, but less than mouse and Macaca microsomes (Figure 2).

More recently, Gerdemann et al. [6] compared the disposition of AFB1 in primary cultures of isolated hepatocytes obtained from mouse, rat, and human liver, using LC-MS-MS to characterize individual biotransformation products (Figure 3).

An important advantage of using intact hepatocytes, versus microsomal or post-mitochondrial fractions of the liver, is that both cytosolic and microsomal enzymes are present and organized in a manner similar to what occurs in vivo. However, a disadvantage is that gene expression can change substantially during the isolation process and over the time course of experimentation. Thus, primary hepatocytes may not accurately reflect the quantitative distribution of specific biotransformation enzymes seen in vivo, including enzymes involved in the biotransformation of AFB1 [20,21]. It is evident from Figure 3 that there are large differences in metabolite distribution across the three species, and that the ratios of different metabolites vary substantially when comparing a low (1 μM) to relatively higher (10 μM) initial AFB1 concentration. Mouse hepatocytes demonstrated nearly complete (99%) biotransformation of the added AFB1 at 1 μM, whereas human hepatocytes metabolized only about 20% of the added AFB1. Rats eliminated 78% of the administered dose over the 4 h incubation. However, at 10 μM AFB1, the intrinsic clearance increased from 19% to 32% in human hepatocytes. Other important species differences in AFB1 metabolites found in this and other studies are discussed in the following sections.

#### Role of Specific Isoforms of CYPs in Activating AFB1 to AFBO, by Species

Humans

The role of specific human CYPs in biotransformation of AFB1 is well established. Human hCYP1A2, hCYP3A4, hCYP3A5, hCYP3A7 and hCYP2A13 are all capable of activating AFB1 to AFB-8,9-epoxide [18,22,23,24,25], but the kinetic characteristics are distinct and important [23]. hCYP3A5 is variably expressed at relatively low levels in the human liver. CYP3A7 is primarily a fetal form that has been shown to activate AFB1 to the 8,9-epoxide [26,27] and, thus, could play an important role in AFB1 disposition if exposure occurs in utero [28]. Aflatoxin–albumin adducts have also been identified in cord blood of newborns whose mothers were exposed to aflatoxin in the diet in The Gambia [29].

In contrast to CYP3A4, which preferentially forms AFQ1, CYP3A5 metabolizes aflatoxin B1 mainly to the exo-8,9 epoxide but is ~100-fold less efficient compared to the formation of AFQ1 [30]. There is a large genetically determined variability in expression of hepatic CYP3A5 between individuals, with many individuals showing no expression. For example, ~40% of African Americans do not express this enzyme [31]. Therefore, differences in expression of CYP3A5 could influence susceptibility to aflatoxins.

hCYP2A13 is expressed in human lung tissue, but not liver. Thus, it is generally agreed that hCYPs 1A2 and 3A4 are the predominant CYP enzymes that can generate AFBO in human liver. hCYP1A2 forms roughly equal amounts of both endo- and exo-AFB-8,9-epoxide, whereas hCYP3A4 appears to form predominantly the exo form of epoxide [18] (and relatively greater amounts of AFQ1). This is important, because AFB1 exo-8,9-epoxide reacts with DNA to give adducts in high yield (>98%). This interaction is characterized by a *K*_d_ of ~1.4 μM, intercalation between base pairs, and rapid reaction to form AFB-N7-guanine adduct [32]. Although some studies suggested that hCYP3A4 is the predominant hCYP in forming AFBO in vitro [33], these and other studies have utilized relatively high concentrations (e.g., >50 μM) of AFB. However, Gallagher et al. [23] examined the reaction kinetics of AFB1 oxidation by hCYP1A2 and hCYP3A4 in both human liver microsomes (HLMs) and in human CYP3A4 and CYP1A2 cDNA-expressed lymphoblastoid microsomes using relatively low concentrations. AFBO formation by cDNA-expressed human CYP1A2 followed Michaelis–Menten kinetics, with a Km of 41 µM and a Vmax = 2.63 nmol/min/nmol P450. In contrast, hCYP3A4 formation of AFBO in cDNA-expressed hCYP3A4 microsomes was sigmoidal and exhibited the kinetic characteristics of “substrate activation”. Application of a sigmoid Vmax model equivalent to the Hill equation produced excellent fits to the cDNA-expressed CYP3A4 data and to the data from HLMs pretreated with the specific hCYP1A2 inhibitor, furafylline, which selectively inhibits hCYP1A2 activity. The Hill model predicted that two substrate binding sites are involved in hCYP3A4-mediated AFB1 catalysis and that the average affinity of AFB1 for the two sites was 140–180 µM, about 4 times higher than the Km for hCYP1A2. However, the hCYP3A4 Vmax values for AFBO formation were substantially greater than that for hCYP1A2, reflective of the relative amounts of the two proteins in human liver microsomes [23].

Using the derived kinetic parameters for hCYP1A2 and h3A4 to model the in vitro rates of AFB1 activation at low substrate concentrations, Gallagher et al. [23] predicted that CYP1A2 contributes to over 95% of AFB1 activation in human liver microsomes at 0.1 µM AFB. However, at in vitro concentrations of AFB1 greater than about 10 µM, CYP3A4 appears to the dominant contributor to AFBO formation. Kamdem et al. [24] conducted a similar kinetic study with baculovirus-expressed hCYP1A2, 3A4, 3A5, and 3A7, as well as human liver microsomes, and concluded that CYP3A4 was the dominant human CYP that forms exo-AFBO. They disagreed with Gallagher et al., stating that “*However, the critically important hepatic expression levels of P450 1A2 and P450 3A4 were not considered in the calculation. The postulated dominant role of P450 1A2 in AFBO production was also not supported by the Vmax of AFBO production, which was highest in the liver with the highest P450 3A4 expression and not in the liver with the highest P450 1A2 expression, as would be expected (in Gallagher et al.* [23]*)*”.

But Kamdem et al. [24] used only five concentrations of AFB1, ranging from 25 to 500 µM, to assess their kinetics, and at these relatively high concentrations were not able to observe the important Hill kinetics characteristics that occur when an enzyme has more than one substrate binding site, one of which serves as a “self-activator” of activity, as shown by Gallagher et al. for AFB1 [23]. Gallagher et al. used nine different substrate concentrations, ranging from 0.5 to 512 µM, which was sufficient to demonstrate non-linearity at low doses, consistent with Hill, but not Michaelis–Menten, kinetics. Numerous other studies have demonstrated non-linear Hill kinetics for hCYP3A4 with numerous substrates [34,35,36,37,38]. Hill kinetics dictate that, at low substrate concentrations (<1 µM for AFB1), the high apparent Km (low binding affinity) for AFB1 precludes any significant catalytic activity by hCYP3A4, even if there are relatively large amounts (high Vmax) of protein available. And Gallagher et al. did consider the relative amounts of CYP1A2 and 3A4, as they used human liver microsomes, which have substantially greater amounts of CYP3A4 protein compared to CYP1A2. Kamdem et al. [24] did not include any substrate concentrations below 25 µM in their kinetic assessment, and thus did not observe Hill kinetics, versus Michaelis–Menten, in their evaluation. This led them to conclude that CYP3A4 was far more important in exo-AFBO formation than CYP1A2, even at low substrate concentrations.

Most importantly the role of hCYP1A2 in the hepatic in vitro activation of AFB1 at low substrate concentrations (0.13 µM) was supported by DNA binding studies using human liver microsomes in the Gallagher et al. study. AFB1-DNA binding in control HLMs (reflecting the contributions of both CYP1A2 and CYP3A4) and furafylline-pretreated microsomes (reflecting the contribution of CYP3A4 only) catalyzed the binding of 1.71 and 0.085 pmol equivalents of AFB1 to DNA, respectively, demonstrating that hCYP1A2 was responsible for 95% of AFB1-DNA adduct formation, even though the amount of CYP3A4 protein was much higher than that of CYP1A2.

The results of Gerdemann et al. [6] from human hepatocytes also illustrate the important kinetic differences in AFBO formation between high and low concentrations of AFB1 and the relative contribution of hCYP1A2 and hCYP3A4 to oxidative metabolism of AFB1 (Figure 3, human). At 1 µM AFB1, AFM1 represents 57% of total metabolites, and AFQ1 is less than 1% of epoxide formation (sum of dihydrodiol, AFB-GSH, and AFB-Lys). However, at 10 µM, AFM1 drops to 11%, AFQ1 increases to 22%, and AFBO increases to 66%. This change in metabolite distribution, and the overall increase in the rate of total biotransformation, is consistent with Hill kinetics for CYP3A4 formation of AFBO and AFQ1. The 10 µM concentration of AFB1 in the hepatocytes was sufficient to induce substrate activation of CYP3A4. Thus, at a low AFB1 concentration of 1 µM, CYP1A2 dominates biotransformation, resulting in large amounts of both AFM1 and AFBO, whereas at a high substrate concentration (10 µM), CYP3A4 is allosterically activated and contributes substantially to overall clearance and increases the amount of both AFBO and AFQ1 formation.

Thus, the studies of Gallagher et al. [23,39,40] and Gerdemann et al. [6] demonstrate that CYP1A2 dominates the activation of AFB1 in human liver microsomes in vitro at sub-micromolar concentrations and support the hypothesis that hCYP1A2 is the predominant CYP responsible for AFBO activation (exo-AFBO) in human liver in vivo at the relatively low dietary concentrations encountered in the human diet, even in high-AFB1-exposure regions of the world.

Finally, it is worth noting that CYP1A2 is a hepatic-specific CYP, consistent with the relatively selective toxicity and carcinogenicity of AFB1 in the liver. However, inhalation exposure to aflatoxin can occur and could potentially cause lung toxicity or cancer if there were CYPs in the lung capable of activating AFB1 to AFBO. Indeed, several studies have demonstrated this potential. Putt et al. [41] demonstrated that rabbit and rat nasal mucosa microsomes were highly active in forming AFBO from AFB1, and demonstrated that cDNA-expressed rabbit CYP2A10 and 2A11, which are expressed in nasal mucosa, were effective at forming AFB-DNA adducts when incubated with AFB1. Several studies have found that human CYP2A6 does have AFBO formation activity, although this is less active than that of either CYP1A2 or 3A4 in cDNA expression assays for AFBO-generating activity [22,42]. hCYP3A4 is also expressed in the human lung, albeit at relatively low levels, and has been shown to activate AFB1 in human lung microsomes [43]. A recent review of occupational exposure studies on mycotoxins [44] found nine reports in the literature that have identified AFB1 in dusts in a variety of settings (animal husbandry (three studies), agriculture (four studies), and waste plants (two studies)). However, few studies have examined lung cancer risk among workers exposed to AFB1-contaminated dust. One small occupational epidemiology study of workers in a plant that extracted oil from linseed and peanuts during the period 1961–1968 assessed the potential for inhalation exposure to AFB1 to cause cancer, including lung cancer [45]. Aflatoxins were measured in the plant, and estimates of exposure were based on inhalation of dust particles containing trace amounts of AFB1. Although the study was small (N = 71), there were seven cases of lung cancer identified among the AFB1-exposed workers, with only three in a matched control group (N = 68). But information was not available on individual smoking histories, so no strong conclusions can be drawn from this small cohort study. No liver cancers were detected in the small cohort.

Since CYP3A4 is expressed in human intestinal tissue, including colon [46,47], and is capable of activating AFB1 to exo-AFBO, it is possible that relatively high concentrations of AFB1 in the diet could induce DNA damage and possibly cancer in the human intestinal tract. Harrison et al. [48] found AFB1-DNA adducts in human colon tissues, and hypothesized that AFB1 could contribute to colon cancer risk. However, to date there is little evidence demonstrating an association between dietary aflatoxin exposure and colon cancer.

b.Non-human primates

There is relatively little information on the role of specific CYPs in the biotransformation of AFB1 by non-human primates. One study [49] compared the microsomal oxidation of AFB1 to AFBO, AFM1, AFP1 and AFQ1 in human liver with two species of non-human primates (*Macaca* and Marmoset). The mean activation of AFB1 to AFBO in human, Macaque, and Marmoset liver microsomes (N = 3 for each) was 68.7, 55.2, and 25.3 pmol/min/mg microsomal protein, respectively, at an AFB1 concentration of 16 µM, with a 2.8-fold range of activity in the three human liver samples. However, an earlier study that compared AFB1 metabolism in hepatic microsomal fractions from human, rat, mouse, and non-human primates (*Macaca nemestrina*) found that, at a relatively high substrate concentration of 124 µM, Macaque liver microsomal activity of AFBO formation was about 3 times higher than that of human liver microsomes (Figure 2; [16]). Thus, it appears that the livers of these two species of non-human primates have a roughly similar AFB1 activation potential as human liver, although there are no studies evaluating which specific CYP enzymes are responsible, nor any studies assessing relative activities at lower substrate concentrations relevant to human exposures (e.g., <0.1 µM).

c.Rats

In 1991 Ch’ih et al. [50] conducted an experiment with isolated rat hepatocytes to evaluate the role of various biotransformation pathways in the overall disposition of AFB1 and determine whether AFB1 metabolites are excreted into the extracellular environment. Cells were incubated with 1500 pmol/10^8^ cells for 60 min, and then the cells and the media were analyzed for the presence of various AFB1 metabolites (Table 1).

The key points here are as follows: (1) ~95% of the AFB1 was metabolized; (2) of that, 83% was biotransformed to AFBO (sum of “bound” and AFB-GSH), and (3) only 1% of the overall oxidative metabolism was attributed to hydroxylated forms (AFM1, AFP1, and AFQ1); (4) of those, AFP1 was the dominant form in rat liver hepatocytes; and (5) there was a remarkable capacity of liver cells to transport metabolites out of the cells and into the extracellular media.

The results of Gerdemann et al. (Figure 3; [6]) in primary cultures of isolated rat hepatocytes are consistent with this finding, showing AFBO formation of about 60% of total biotransformation products. These results also suggest that rat CYP3A2 may exhibit similar allosteric activation as human CYP3A4, since AFQ1 was less than 1% of overall metabolites at 1 µM but increased to 20% of overall metabolites at 10 µM.

Perhaps surprisingly, few studies have evaluated which specific rat CYPs are involved in AFBO formation. Although the specific rat CYPs involved in AFBO formation are not known for certain, there is some evidence suggesting that rCYP3A2 (orthologous to hCYP3A4) and rCYP2C11 are involved in metabolic activation of AFB1 in the rat [51,52]. In contrast to humans, rat CYP1A2 appears to play little role in hepatic activation of AFB1 [53], and it is likely that rCYP3A2 is the predominant CYP in rats that forms AFBO.

d.Mice

As illustrated in Figure 2, mouse liver microsomes have relatively high activity to form AFBO from AFB1, nearly twice that of rat and about 5 times greater than human liver microsomes. As noted previously, adult mice are highly resistant to the carcinogenic effects of AFB1, but apparently it is not because they cannot form the genotoxic AFBO. As with rats, there is relatively little information on which specific CYPs form AFBO in mouse liver, although mouse Cyp3a11 and 3a13 have been shown to possess AFBO-forming activity [54,55]. However, there is some evidence that mouse Cyp2a5 may also be involved in AFB1 activation [56]. The role of mouse Cyp1 enzymes in AFB1 is unclear. However, one study found that inhibitory antibodies to mouse Cyp1a2 had less inhibitory effect on AFB1 activation than did inhibitory antibodies to mCyp2b or mCyp3a, suggesting that mCyp1a2 may play only a minor role in AFB1 activation in mouse liver [57]. Thus, it remains uncertain exactly which mouse Cyp enzyme(s) is/are responsible for activation of AFB1 to AFBO, although the Cyp3 family is likely to contribute significantly.

e.Fish/salmonids

The sensitivity of rainbow trout to hepatocellular carcinoma induction by AFB1 is due in large part to trout CYP2K1, a constitutively expressed CYP in liver [58]. Trout CYP2K1 (previously designated as trout liver LM2) was purified from liver of BNF-treated trout [59,60]. CYP2K1, down-regulated by BNF treatment, could be distinguished from the BNF-induced CYP1A1 (previously designated as trout liver LM4b) by its *lambda* max of 450 nm in the CO-reduced versus CO spectra compared to LM4, and by immunochemical specificity with rabbit antibodies raised to either CYP [60,61]. The isozymes could also be distinguished by substrate specificity. CYP2K1 did not display any activity in vitro toward benzo[*a*]pyrene but exhibited high activity toward AFB1 [58,60]. Conversely, trout CYP1A1 had only 15% of the turnover with AFB1 compared to CYP2K1 and failed to produce any adducts with added calf thymus DNA.

Purified trout CYP2K1 was incubated with ^14^C-AFB1 and metabolites separated by HPLC and quantified by scintillation counting of peaks (365 nm) corresponding to AFB1-8,9-diol (hydrolysis product of the epoxide), AFQ1, and AFM1 (Figure 1). CYP2K1 exhibited high regioselectivity (72% of metabolites produced) toward production of the 8,9-*exo*-epoxide (Table 2). In addition to AFB1, purified hepatic CYP2K1 and BV-CYP2K1 (expressed with baculovirus) exhibit regioselectivity toward omega-1 hydroxylation of lauric acid [62,63,64]. AFB1 is also epoxygenated to produce covalent DNA adducts in vivo in zebrafish [65]. Although zebrafish are only 4-fold less efficient than trout at AFB1-DNA adduction in vivo, they exhibit relative resistance to AFB1-induced HCC [66]. CYP2K6, expressed from zebrafish, has activity toward conversion of AFB1 to the exo-8,9-epoxide but, unlike trout CYP2K1, has no activity toward lauric acid [67].

In addition to a high rate of CYP2K1 conversion to the AFB1-exo-epoxide, there is evidence in trout of a slow rate of inactivation of the epoxide by liver GST or trapping with other nucleophiles [68]. This factor, plus the slow rate of DNA repair of bulky adducts such as the 8,9-dihydro-8-(N7-guanyl)-9-hydroxyaflatoxin B1, in salmonids enhances the carcinogenic potency of AFB1 [69,70]. The species difference in salmonids (rainbow trout and coho salmon) was attributed solely to the rate of AFB1-8,9-epoxide formation in trout [70]. A log-linear relationship, evident at exposure levels of 0.05–100 µg/kg, between AFB1 dietary dose in trout and HCC, was demonstrated in a 40,000 animal ED001 (effective dose producing 0.1% incidence) tumor study [66]. A log-linear response between AFB1 dose and liver DNA adduction has been demonstrated both in trout and rat [71].

f.Avians (Turkeys, Chickens, Ducks)

Avians, especially turkeys, are extremely sensitive to the toxic effects of AFB1 [72,73,74]. As discussed in the Introduction, this extreme sensitivity to AFB1 was graphically demonstrated when it was discovered to be the etiological agent of “Turkey X Disease”, resulting in widespread deaths of turkeys and other poultry throughout Europe in the 1960s. This event was determined to have been caused by AFB1-contaminated Brazilian peanut meal used for poultry feed [75]. Among avian species, turkeys are considered to be the most susceptible to aflatoxins, quails have intermediate susceptibility, while chickens are considered relatively resistant [72]. Wild turkeys appear to be less susceptible to AFB1 than their domesticated counterparts [76].

In contrast to the abundant data on the role of individual mammalian P450s on AFB1 bioactivation, there is limited information on avian orthologues. Reflecting whole-animal susceptibility differences, turkey liver microsomal AFB1 epoxidation activity was determined to follow a rank order of turkey > quail > chicken [77]. Using P450-orthologue specific substrates, inhibitors, and antisera, CYP1A2, and, to a lesser extent, 3A4, orthologues were postulated to be critical in AFB1 bioactivation in turkey liver [73]. Microsomal AFB1 bioactivation in turkeys is age-related, in that younger birds are more efficient than older birds [78]. The critical role of CYP1A5 in the biological action of AFB1 was further supported by a demonstration that inhibitors of this enzyme alleviate symptoms of aflatoxicosis in poultry exposed to dietary AFB1. Dietary butylated hydroxytoluene (BHT) protects against AFB1-associated symptoms in turkeys, in part, due to its inhibitory effect on P450 1A5, exhibiting Michaelis–Menten competitive inhibition kinetics (K_i_ = 0.81 μM) [79]. However, these early microsomal metabolic studies were of limited value in identifying, with certainty, AFB1-bioactivating P450 orthologues in avian liver because they relied upon substrates, inhibitors, and antisera validated in mammalian rather than in avian systems.

Confirmation of the central role of avian P450s 1A2 and 3A4 orthologues in AFB1 bioactivation was provided when turkey P450s were cloned and functionally characterized [80]. Turkey liver CYP1A5, the first functional protein amplified from turkeys to be cloned, was predicted to contain 528 amino acids with 94.7% sequence identity to chicken CYP1A5, the structure of which was equivalent to that of human CYP1A1, with seven exons and six intervening sequences. Intron sequences were highly similar to the available chicken sequences (Introns 1 and 4–6), with similarity values of aligned intron and UTR sequences exceeding 91% [80]. Like its human homologue, the *E. coli*-expressed CYP1A5 efficiently bioactivated AFB1 to AFBO, and in addition, produced AFM1. The CYP1A family identity of CYP1A5 was confirmed using specific inhibitors of catalytic activities, and through genetic mapping showing that the structure of the turkey gene was shown to be equivalent to that of the human *CYP1A* genes with seven exons of 38, 858, 127, 90, 124, 87, and 307 bp, and six introns. A single nucleotide polymorphism (SNP) in the 3′ UTR (untranslated region) was used to assign *CYP1A5* to turkey linkage group M16 (equivalent to chicken chromosome 10) [81]. Heterologously expressed CYP1A5 is extremely efficient in AFB1 bioactivation to exo-AFB1 as demonstrated by kinetic values of Km and Vmax of exo-AFBO formation of 65 ± 12 μM and 0.61 ± 0.037 nmol/min/nmol P450, respectively (R^2^ = 0.93). Kinetics of AFM1 formation were 34 ± 9 μM and 0.91 ± 0.070 nmol/min/nmol P450, respectively [80].

In a follow-up study, a CYP3A4 orthologue in turkey liver was cloned, heterologously expressed, and functionally characterized, with particular focus on its activity toward AFBO formation. The CYP3A37 gene has an open reading frame (ORF) of 1512 bp, and the protein is predicted to be 504 amino acids with 97% identity to chicken P450 3A37 [82]. These studies showed that the turkey 3A37 was significantly less efficient in AFB1 bioactivation than turkey 1A5, as exemplified by Km and Vmax values for the formation of exo-AFBO of 287 ± 21 μM and 1.45 ± 0.07 nmol/min/nmol P450, respectively [83]. Sequencing of the turkey genome in 2010 enabled a precise comparative assignment of the relative roles of turkey CYPs 1A5 and 3A37 in AFB1 bioactivation [84]. Building on this DNA sequence information, Rawal and Coulombe [85] constructed anti-peptide antibodies directed against P450 1A5 and 3A37 as tools to investigate the role of these enzymes in bioactivation of AFB1 in turkey liver microsomes, especially at relatively low concentrations likely to be achieved in turkey liver in vivo. Anti-peptide antisera are valuable tools for isolating the catalytic contribution of individual AFB1-metabolizing P450s in multi-enzyme systems such as liver microsomes to identify the predominant enzyme. Metabolism in immuno-inhibited microsomes resembled that of individual enzymes. For example, when the contribution of P450 3A37 was immuno-inhibited, microsomes oxidized AFB1 to only two detectable products, exo-AFBO and AFM1, the kinetics of which were characterized by hyperbolic Michaelis–Menten kinetics of tCYP1A5. In incubations where 1A5 was inhibited, 3A37 produced exo-AFBO and AFQ1 following Hill kinetics. At 0.1 μM AFB1, (approximating AFB1 concentrations in livers of exposed animals), 1A5 contributed ~98% of the total exo-AFBO formation (Figure 1). At this concentration, 1A5 accounted for a higher activation: detoxification (50:1, exo-AFBO: AFM1) compared to 3A37 (0.15:1, exo-AFBO:AFQ1), suggesting that 1A5 is high, while 3A4 is the low-affinity enzyme in turkey liver. AFB1 bioactivation by 3A37 at low AFB1 concentrations is minimal. The data support the conclusion that, in turkey liver, CYP1A5 is the dominant enzyme responsible for AFB1 bioactivation and metabolism at environmentally relevant AFB1 concentrations in turkey liver [85]. The CYP3A37 enzyme predominates in AFBO formation but only at high, pharmacologically irrelevant AFB1 (i.e., >100 μM) concentrations (Figure 4).

### 2.2. Step #2 Oxidative Biotransformation of AFB1 to Aflatoxin M1

In 1963 Allcroft et al. [86,87] proposed that milk from dairy cattle fed AFB-contaminated feed could contain aflatoxin or a toxic metabolite derived from it. Ducklings administered milk from dairy cattle fed aflatoxin-contaminated feed demonstrated toxic symptoms similar to that seen when the ducklings were fed contaminated groundnut meal. Soon after, De Iongh et al. identified the toxin in milk by thin layer chromatography, and proposed calling it Aflatoxin M1 [88]. It is a simple hydroxylation product of AFB1, in the 9a position (Figure 1, reaction #2).

AFM1 is formed in humans following ingestion of dietary AFB1, as indicated by early studies in the Groopman laboratory, where AFM1 was one of several AFB1 metabolites found in human urine by affinity chromatography [17].

Although generally considerably less toxic and carcinogenic than AFB1, AFM1 can be activated to the 8,9-epoxide (AFMO) and thus exhibits similar toxicity to AFB1. The relative carcinogenic potency of AFM1 in Fisher rats is about 2–10% of that of AFB1 [89]. However, Purchase et al. [90] found that AFM1 had similar acute toxicity as AFB1 in 1-day old ducklings. Thus, species differences in the relative toxicity of AFM1 and AFB1 could be important. Because milk is a food staple that can constitute 100% of the diet of infants, the FDA tolerance level for AFM1 in milk is 0.5 ppb, 40 times lower than the tolerance for AFB1 in food (20 ppb) (See: https://www.fda.gov/regulatory-information/search-fda-guidance-documents/guidance-industry-action-levels-poisonous-or-deleterious-substances-human-food-and-animal-feed#afla (accessed on 28 December 2024).

Numerous studies in several species have investigated the carcinogenic effects of AFM1. A comparison of AFB1 with AFM1 in weanling Fisher rats given 25 mg of either AFB1 or AFM1 by gavage for 5 days per week for 8 consecutive weeks found that 100% of the rats given AFB1 had liver tumors at the termination of the experiment, whereas only 1 (3%) rat in the AFM1 group had a liver tumor, although many had preneoplastic lesions [4]. Cullen et al. [91] also compared the relative carcinogenic potency of AFM1 with AFB1 in Fisher rats in a chronic bioassay, and found AFM1 to be carcinogenic in both the liver and intestine, with a potency about 2–10% less than AFB1. AFM1 was also shown to be a potent carcinogen in rainbow trout [92]. Trout hepatocytes exhibit significantly lower DNA adduction in vitro when incubated with AFM1 compared to AFB1 [93]. Hepatic DNA adduction in vivo following injection of trout fry with AFM1 was 11% that of AFB1 and the yield of HCC at 12 months was 8.6% that of AFB1 [94].

As noted above, although AFM1 is often considered a “detoxification” product, it can also undergo activation to the reactive epoxide (AFMO). Studies with human microsomes and lymphoblastoid cells (cell line MCL-5) that were stably transfected with human cDNAs for CYPs 1A2, 2A6, 3A4, and 2E1 together with microsomal epoxide hydrolase were used to evaluate the formation of metabolites of AFM1 [95]. Overall, epoxidation of AFM1 by human liver microsomes was substantially less than seen for AFB1, although some AFM1-dihydrodiol and AFM1-GSH conjugates were evident.

#### Role of Specific CYP Isoforms in Activating AFB1 to AFM1, by Species

Humans

AFM1 was the major AFB1 metabolite in primary cultures of isolated human hepatocytes at a relatively low substrate concentration of 1 µM (Figure 3; [6]), accounting for 57% of overall metabolites.

hCYP1A1 and hCYP1A2 both form AFM1 from AFB1 [18,23,39]. hCYP1A1 forms predominately AFM1, whereas hCYP1A2 forms both AFM1 and AFBO in a ratio of AFM1:AFBO of 1:2.5 [39]. hCYPIA2 produced a higher ratio of activation/ deactivation products (AFBO:AFM1 = 3:1), and the rate of formation of AFBO was about 30-fold higher than that of hCYP1A1. In contrast, hCYP1A1cDNA-expressed microsomes produced >10 times more AFM1 than AFBO [39]. However, hCYP1A1 is not constitutively expressed in human liver, although exposure to AhR ligands found in cigarette smoke and the diet may cause some induction of hCYP1A1 in human liver.

Although human microsomes can form AFMO from AFM1, it is not clear which specific human CYPs are capable of activating AFM1 to the DNA-binding epoxide, or the protein-binding dihydrodiol [95].

Urinary AFM1 has been used as a biomarker of aflatoxin exposure in numerous studies [96,97,98,99,100,101,102,103].

b.Non-human primates

Macaca liver microsomes have a relatively low initial rate of formation of AFM1, similar to humans (Figure 2; [16]. The ratio of AFM1:AFBO in Macaca is 1:10, whereas in humans it is 1:3, largely because the Macaca microsomes are approximately 3 times more active than human microsomes in forming AFBO (Figure 2). Bammler et al. [49] found that liver microsomes from the Marmoset monkey also had a ratio of AFM1:AFBO of about 1:10.

c.Rats

The data in Figure 2 demonstrate that rat liver microsomes are somewhat more efficient at forming AFM1, compared to those of human liver, but the ratio of AFBO:AFM1 is about the same. In isolated rat hepatocytes, AFM1 constituted ~3% of total AFB1 metabolites (Table 1) [50]. However, Gerdemann et al. found that AFM1 accounted for 38% of total AFB1 metabolites after 4 hr of incubation (Figure 3; [6]). Although few, if any, studies have explicitly evaluated specific rodent CYPs for AFM1 formation, it is likely that rat CYP1A isoforms are responsible, as has been described in humans. Gurtoo et al. [104] demonstrated that induction of CYP enzymes in rat liver by the AhR agonist β-naphthoflavone (BNF) substantially increased the formation of AFM1, but also reduced hepatocarcinogenicity of AFB1 by 75%. This is consistent with AFM1 formation via CYP1A1 activity in the rat [39].

d.Mice

Mouse liver microsomes are relatively efficient at forming AFM1, compared to either rat or human microsomes (Figure 2). In primary cultures of mouse hepatocytes at a low substrate concentration of 1 µM, AFM1 accounted for ~3% of total metabolites. However, this increased to 19% at 10 µM (Figure 3; [6]). As with rats, there is no specific Cyp mouse form that has been demonstrated to do this, but it is a reasonable assumption that AFM1 is formed by a mCyp1a isoform. There was little strain differences in formation of AFM1, among eight different strains of mice examined, with the exception of the CD-1 mouse, which had activity about 50% of the activity seen in the other seven strains [105].

e.Fish/Salmonids

Treatment of trout with CYP1A1 inducers such as BNF, PCBs, or indole-3-carbinol prior to or concurrent with AFB1 dosing resulted in a significant reduction in hepatic DNA adduction and subsequent reduced incidence and multiplicity of HCC. Consistent with these results, the metabolic profile of AFB1 both in vitro and in vivo is altered by treatment with BNF, resulting in increased production of AFM1 and a marked reduction in AFL1 [58,69,106,107].

f.Avians (Turkeys, Ducks, Chickens)

As mentioned above, recombinant *E. coli*-expressed CYP1A5, like the human orthologue 1A2, produces AFM1, in addition to efficiently bioactivating AFB1 to AFBO [78]. Otherwise, scant comprehensive data exist on metabolic products of AFB1 in the livers of avian species. But reports of residues of AFB1 metabolites in the tissues of AFB-challenged birds have been reported. For example, in AFB1-fed quail, residues of AFM1, AFB2α, and AFL were detected in eggs [108]. Similarly, AFL and AFM1 was measured in the liver, kidney, and thigh of both male broilers and hens fed AFB1 [109]. Breast meat from laying hens fed AFB1 contained AFL and AFM1 [110].

### 2.3. Step #3 Oxidative Biotransformation of AFB1 to Aflatoxin P1

Aflatoxin P1 (AFP1) is formed via oxidative demethylation (Figure 1, reaction #3). There appears to be substantial differences among species in the extent of O-demethylation of AFB1 to form AFP1. Based on hepatic microsomal assays, AFP1 was not detected in monkey or human liver microsomes, but was a substantial metabolite produced by rat liver microsomes (Figure 2), and constituted 13% of total oxidative metabolites in isolated rat hepatocytes, of which 93% was found as the glucuronide conjugate, and nearly all of that was exported out of the hepatocytes (Table 1; [50]). Consistent with that observation, the glucuronide conjugate of AFP1 is also a major metabolite of AFB1 in rat bile [111]. However, comparative studies in primary cultures of mouse, rat, and human hepatocytes found that AFP1 was the major oxidative metabolite in mouse hepatocytes, but was a very minor metabolite in rat hepatocytes, and was largely undetectable in human hepatocytes (Figure 3; [6]). Bammler et al. [49] also found no measurable AFP1 formed in microsomal incubations from human, Macaque, and Marmoset monkey livers (N = 3 for each), at 16 μM AFB1 concentration.

Groopman et al. [112] found AFP1 in the urine of human subjects exposed to AFB1 in the diet, although there was no significant correlation between urinary AFP1 levels and estimated dietary intake. In contrast, highly significant correlations were seen between dietary exposure and AFM1 and AFB-N7-Guanine adducts in urine. The authors noted that biliary excretion may be the primary route of elimination of AFP1, and that unconjugated AFP1 in the urine may be highly variable depending on the dose and extent of conjugation, making urinary AFP1 an unreliable biomarker of exposure.

It should be noted that the microsomal enzyme assays typically used to measure AFB1 oxidation may fail to capture the true extent of AFP1 formation. It is possible that AFP1 is rapidly conjugated with glucuronic acid, since UDP-glucuronosyl transferases (UGTs) are also microsomal enzymes and specific membrane transport processes in the endoplasmic reticulum concentrate potential substrates in microsomal membranes [113]. If AFP1 was conjugated after formation in the microsomes, it would not be detected in the HPLC assays used to identify unconjugated AFP1. However, conjugation requires the presence of a significant concentration of uridine diphosphoglucuronic acid (UDPGA), and it is unknown whether residual levels of UDPGA may be present in incubations following isolation of the microsomal fraction.

#### Role of Specific Isoforms of CYPs in Activating AFB1 to AFP1, by Species

Humans

There is relatively little information on which specific CYPs are responsible for the O-demethylation of AFB1 to AFP1 in human liver. Although structurally different than AFB1, the plant toxin aristolochic acid (AA) undergoes a similar O-demethylation of reaction of an anthracene-based O-methyl analog in AA. Stiborova et al. [114] demonstrated that multiple human CYPs contributed to the O-demethylation of AA, but that CYP1A2 was responsible for approximately half of the overall activity, whereas CYP3A4 contributed about 10% of overall activity. Thus, it is reasonable to hypothesize that CYP1A2 contributes to the formation of AFP1 as well as AFBO and AFM1.

b.Non-human primates

As noted above, Bammler et al. [49] compared hepatic microsomal biotransformation of AFB1 in two non-human primate species, *Macaca nemestrina* (Macaque) and *Callithrix jacchus* (Marmoset), with human liver microsomes, and AFP1 formation was below the limit of detection in all six monkey livers (N = 3 for Macaque and Marmoset), as well as in all three human liver microsomal preparations.

No studies were identified that have evaluated specific monkey CYPs for formation of AFP1.

c.Rats

AFP1 is a major oxidative metabolite of AFB1 in rat liver microsomes, as shown in Figure 2. As noted above, Ch’ih et al. [50] also demonstrated that AFP1 is, by far, the dominant hydroxylated AFB1 metabolite formed in isolated rat hepatocytes (Table 1), although Gerdemann et al. [6] found AFP1 to be a minor metabolite in isolated rat hepatocytes. It is unclear whether methodological, and/or rat strain, differences are responsible for this apparent discrepancy. To date, there are no studies that have definitively identified which rat CYP(s) is/are responsible for the formation of AFP1 in the liver.

AFP1 can also undergo a second oxidation reaction for form 4,9a-dihydroxy-AFB1. The same product can be formed via secondary oxidation of AFM1 and appears to be conjugated with glucuronic acid [115].

d.Mice

AFP1 is formed at levels substantially greater than AFQ1 or AFM1 in mouse liver microsomes (Figure 2) and isolated hepatocytes (Figure 3), although, as with rats, there is little information on the specific Cyp form(s) in the mouse that is/are responsible for AFP1 formation.

e.Fish/salmonids

AFB1 O-demethylation to AFP1 is a minor or non-detectable metabolite in trout both in vitro and in vivo. AFP1 has been shown to be a much weaker mutagen and carcinogen in trout [116].

f.Avians (Turkeys, Chickens, Ducks)

AFP1 has not been identified as a significant oxidative metabolite of AFB1 in turkeys or other avian species.

### 2.4. Step #4 Oxidative Biotransformation of AFB1 to Aflatoxin Q1

Hydroxylation of AFB1 on the cyclopentanone ring gives rise to AFQ1 (Figure 1, reaction #4). In contrast to AFM1, there is little information to suggest that AFQ1 exhibits any significant toxicity, and thus is generally considered a complete detoxification product. The acute toxicity of AFQ1 is only about 5% of AFB1, and in vitro mutagenicity assays found little mutagenic activity for AFQ1 [117,118]. AFQ1 is the predominant in vitro metabolite of AFB1 in human and *Macaque* (monkey) liver microsomes, but is a relatively minor metabolite in rodent (rat, mouse) microsomes (Figure 1) [16,119,120].

#### Role of Specific Isoforms of CYPs in Activating AFB1 to AFQ1, by Species

Humans

There is substantial evidence that hCYP3A4 is the predominant, if not sole, form of hCYP responsible for the formation of AFQ1 in vitro [23,24,33,40,120,121,122]. As discussed above for AFBO production by human CYPs, CYP3A4 shows unusual non-linear kinetics for AFQ1 formation as well as for AFBO [23]. But even at relatively low concentrations of AFB1, the relatively high levels of expression of hCYP3A4 in human liver are adequate to form substantial amounts of AFQ1, as suggested by several in vivo studies.

Ezekiel et al. [123] reported that AFQ1 was the predominant aflatoxin metabolite found in urine of infants who were either exclusively breast-fed or non-exclusively breast fed, in a region of the world with notably high mycotoxin exposure (Ogun state, Nigeria). AFQ1 was detected in the urine of 20 out of 23 (87%) exclusively breast-fed infants and 24 of 42 (57%) non-exclusively breast-fed infants. AFM1 was detected in only 6 of the total 65 infant urine samples collected from both cohorts, and the levels of AFM1, where detected, were substantially lower than those of AFQ1. The observation that AFQ1 was present in the urine of infants fed exclusively breast milk demonstrates that AFQ1 was a major metabolite of AFB1 in the nursing mothers and is passed into breast milk. In a study in China, AFQ1 and AFM1 were measured in the urine and feces of 83 young Chinese males selected from a larger population based on the presence of detectable AFM1 in their urine. AFQ1 was measured at levels ~60 and 260 times higher than AFM1 levels in feces and urine, respectively [124].

As noted previously, Gerdemann et al. [6] found AFQ1 was a very minor metabolite at a low (1 µM) AFB1 concentration, but a major metabolite at a higher (10 µM) AFB1 concentration, consistent with the dose-dependent allosteric activation found for AFB1 metabolism to AFBO and AFQ1 by human CYP3A4 [23,39].

b.Non-human primates

Roebuck and Wogan [5] demonstrated that AFQ1 was, by far, the predominant metabolite formed from male rhesus monkey post-mitochondrial supernatant liver preparations (similar to microsomal fractions except that cytosolic enzymes were also present). The formation of AFQ1 in the two monkey liver fractions was higher than that seen for three human liver preparations. Formation of AFQ1 in both rhesus monkey and human liver tissue was >10 times greater than the small quantity of AFQ1 formed in mouse and rat liver fractions. Krieger et al. [118] also demonstrated that AFQ1 was the major metabolite of AFB1 in rhesus monkey liver, with a Vmax about 10 times higher than that for AFM1.

Bammler et al. [49] compared hepatic microsomal biotransformation of AFB1 in two non-human primate species, *Macaca nemestrina* (Macaca) and *Callithrix jacchus* (marmoset), with human liver microsomes, and also found that AFQ1 was, by far, the major oxidative metabolite, with an initial rate of oxidation of ~450 pmol/min/mg protein in both species, 3 times faster than the rate of formation of AFQ1 in human liver microsomes.

No studies using specific CYP enzymes from non-human primates were identified, although it is highly likely that the formation of AFQ1 was mediated by CYP3A64, the rhesus monkey homolog to human CYP3A4, as the deduced amino acid sequence of CYP3A64 is 93% homologous to human CYP3A4, and they have similar activities toward a variety of CYP3A4 substrates [125].

c.Rats

In contrast to primates, rats appear to form less AFQ1 than AFBO (Figure 2), at least at low substrate concentrations (Figure 3; Table 1). In vivo studies support the relatively low rate of AFQ1 formation in rats. For example, Tang et al. [126] administered male F344 rats 250 ug/kg of AFB1 5 days per week for three weeks. Daily urinary AFQ1 levels were 10–50 times lower than AFM1 levels over the 15 days of measurements. No kinetic studies on the oxidative formation of AFQ1 by rat CYP3A2, the ortholog to human CYP3A4, were identified, so it is not clear why rat CYP3A2 (or rat CYP3A1) seems to have much less catalytic activity toward AFQ1 formation, relative to primate CYP3A forms. However, the strong substrate concentration dependence of AFQ1 formation in rat hepatocytes suggests that rat CYP3A2 may follow similar Hill kinetics (allosteric activation), as shown for human CYP3A4 [23,39].

d.Mice

As with rats, mouse liver microsomes seem to have relatively little ability to form AFQ1, as mouse liver microsomes were about 5- and 2-fold less active in forming AFQ1 than human and rat liver microsomes, respectively (Figure 2). No information was identified as to which mouse Cyp(s) is responsible for the relatively small amount of AFQ1 formed, but it would be reasonable to assume that mCyp3a11 was primarily responsible. Cyp1a2 in mouse liver (C57Bl/6 strain) was expressed at about 75% of the level of Cy3a11 [127].

e.Fish/Salmonids

As with mouse and rat, AFQ1 is a minor metabolite in trout, constituting less than 5% of the total metabolites produced by trout liver microsomes [59] (Table 2) and is the weakest mutagen and carcinogen of the primary AFB1 metabolites produced in trout [116].

f.Avians (Turkeys, Ducks, Chickens)

Like human P450 3A4, the *E. coli*-expressed 3A37 biotransformed AFB1 to exo-AFBO and to AFQ1 and possessed nifedipine oxidation activity, both of which were inhibited by the CYP3A4 inhibitor 17α-ethynyl estradiol. The kinetics of oxidation of AFB1 to exo-AFBO followed apparent sigmoidal Hill kinetics, suggestive of an allosteric interaction between the enzyme and the AFB_1_ substrate, similar to that described for hCYP3A4 [23]. The Hill coefficient (*n*) value for tCyp3A37 was 1.9 for exo-AFBO and 1.6 for AFQ1, indicative of positive cooperativity.

## 3. Phase I Biotransformation of AFB1: Reduction Reactions

Reduction reactions are relatively uncommon in vivo, except in the somewhat anerobic environment of the intestine, where the microbiome readily does both oxidation and reduction reactions. For aflatoxin, there is only one known reduction reaction of AFB1.

### Step #5: Reduction of AFB1 to Aflatoxicol (AFL)

Several in vitro studies in a variety of species have demonstrated that the cyclopentanone ring of AFB1 can be reduced to AFL (Figure 1, #5) [128,129,130,131,132]. The reaction is enzymatically mediated via cytosolic reductase(s), but little is known about which specific enzyme(s) is/are involved. Salhab and Edwards [132] compared liver cytosolic fractions from eight species for AFL-forming activity. Rabbit and trout liver formed large quantities of AFL, but the other six species, including human liver cytosol, showed little to no AFL formation. However, once formed, AFL is oxidized back to AFB1 via microsomal dehydrogenases present in the liver. When synthetically prepared AFL was introduced into microsomal fractions from different species (in the presence of carbon monoxide to inhibit CYP oxygenases), liver microsomes from all eight species tested showed some ability to oxidize AFL back to AFB1, with human and hamster liver showing the highest activity [132].

Several studies suggested that AFL was highly toxic to ducklings [133] and mutagenic in different assays [134]. Thus, reduction of AFB1 to AFL may not be a detoxification reaction because it can quickly be oxidized back to AFB1.

Given the relatively small amount of information on the specific reductases and dehydrogenases involved in reduction and oxidation of AFL, we will not detail these reactions by each species. Although early studies suggested that differences in the rate of reductive formation of AFL and its oxidation back to AFB1 may be important in determining species sensitivity [132], very little research on this pathway has been completed in the last 40+ years, with the exception of a 2020 study of poultry liver. Murcia and Diaz [135] evaluated enzyme kinetics for both AFB1 reductase and AFL dehydrogenase activity in 12,000× *g* supernatant fractions of liver from chicken (two different strains), duck, turkey, and quail liver. There were substantial differences in the kinetic characteristics across these four avian species. The authors concluded that “*The ratio AFB1 reductase/AFL dehydrogenase enzyme activity was inversely related to the known in vivo sensitivity to AFB1, being highest for the chicken, lowest for the duck and intermediate for turkeys and quail. Since there is no evidence that AFL is a toxic metabolite of AFB1, these results suggest that AFL production is a detoxication reaction in poultry*”.

## 4. Phase II Biotransformation of AFB1: Hydrolysis, Conjugation and Reduction Reactions

### 4.1. Step #6 Hydrolysis of AFB1 to Aflatoxin B1-Dihydrodiol

Both the exo- and endo- conformations of AFBO are highly unstable and subject to rapid hydrolysis in the presence of water, giving rise to the dihydrodiol product (Figure 5, reaction #6) (reviewed in [136]).

Although chemically synthesized AFBO is rapidly hydrolyzed in an aqueous environment, the endogenous formation of AFBO in the hydrophobic environment of the endoplasmic reticulum by CYP enzymes must be somewhat protected from spontaneous hydrolysis, because glutathione (GSH) conjugation of the epoxide can occur in the presence of GSH and glutathione-S-transferases (GSTs; discussed in detail in the next section). But hydrolysis of AFBO to the dihydrodiol product does occur, and in the absence of significant GST activity, is a major pathway of elimination for AFBO. Indeed, the half-life of exo-AFBO in an aqueous environment was measured to be approximately 1 s [33]. Because of the high lability of AFBO in an aqueous environment, it was often assumed that hydrolysis was non-enzymatic within the cells.

However, some studies have demonstrated that microsomal Epoxide Hydrolase (mEH) may play an important role in AFBO elimination.

#### Role of mEH in Hydrolyzing AFBO to AFB-Dihydrodiol, by Species

Humans

Guengerich et al. [33,136,137] did not observe any significant effect of purified human EH on the recovery of N7-guanyl-DNA adducts in experiments where synthetic AFBO was mixed with DNA. They did, however, observe a slight increase in the rate of hydrolysis of exo-AFBO (from 0.64/s to 0.78/s) when 19 μM mEH was added to the system [33]. They also used an Ames mutagenesis assay (*Salmonella* strain TA1535) that contained cDNA-expressing hCYP3A4 and varying amounts of purified mEH to evaluate whether mEH could reduce AFB1 mutagenesis. They found a very slight reduction in mutagenesis at high ratios of mEH:CYP3A4 in one purified human mEH sample, but little to no effect with a second. They concluded: “*Further studies are needed to evaluate the role of mEH in the metabolism of aflatoxin-8,9-epoxide*” [33].

Kelly et al. [138] examined the potential role of human mEH in hydrolysis of hCYP1A2-generated AFBO in recombinant yeast, by measuring the effects of co-expression of human mEH in AFB-DNA adduct formation (Figure 6).

Co-expression of human mEH with an AFBO-generating system (CYP1A2) resulted in a ~50% decrease in AFB-DNA adducts in the yeast. Insertion of a non-functional mEH cDNA (∆-mEH) had no protective effect. The yeast strain used was developed for mutagenesis assessment via a *Trp*-reversion assay, so mutagenesis assessment was also completed. Similar to the findings for DNA binding, the rate of AFB-induced mutagenesis was also reduced by about 50%. Finally, they also assessed the impact of co-expression of mEH in the Ames *Salmonella* reversion assay, using hCYP1A2-expressing microsomes as an activation system.

The Gerdemann et al. [6] study of AFB1 biotransformation in isolated human hepatocytes also suggested a potentially important role of mEH in the absence of significant GST activity toward AFBO. AFB1–dihydrodiol represented 65% of total AFB1 metabolites at the higher AFB1 concentration of 10 µM (Figure 3). However, it is not possible to distinguish between enzymatic and non-enzymatic hydrolysis in this study.

b.Non-human primates

No studies were identified that examine the putative protective effects of primate mEH in enzymatic hydrolysis of AFBO.

c.Rats

As discussed above for human mEH, Guengerich et al. [33] assessed the ability of purified rat mEH to reduce mutagenicity in the Ames *Salmonella* assay that utilized hCYP3A4 as an activating system, with 20 μM AFB. Rat mEH reduced AFB1 mutagenesis by about 50%, which was substantially larger than the modest mEH protective effect seen in one of two human mEH samples.

d.Mice

No studies were identified that addressed the potential role of murine mEH in detoxifying AFBO. However, because of the very high AFBO-GST activity presence in mouse liver, it is unlikely that mEH plays a significant role in protecting mouse DNA from damage, given the very high protection afforded by mGSTA3-3 (discussed in detail later). This conclusion is supported by the data in Figure 3, where no AFB-dihydrodiol was detected in mouse hepatocytes exposed to either 1 μM and 10 μM AFB1, where 80–90% of total metabolites were either AFB-GSH or AFP1 [6].

e.Fish/Salmonids

Trout liver has been shown to express both cytosolic and microsomal epoxide hydrolase (EH) activity [139]. However, no studies have documented the contribution of microsomal or cytosolic EH in hydrolysis of AFB1-8,9-exo-epoxide, the major metabolite of AFB1 in this species.

f.Avians (Turkeys, Chickens, Ducks)

Diaz and Murcia [140] demonstrated that microsomes prepared from the livers of ducks, turkeys, and chickens produced AFB-dihydrodiol from AFB1. The authors showed that metabolic activity, as measured by Km, showed significant differences among these avian species: ducks were almost 13 times lower that turkey, 20 times lower than quail, and 30 times lower than the chicken breeds [140]. Activity profiles against prototype inhibitors, substrates, and antibody reactivity in immunoblotting implied the significant involvement of CYP2A6 and CYP1A1 orthologues for the production of AFB-dihydrodiol in microsomes prepared from quail and chicken livers.

### 4.2. Step #7 Reduction of AFB1–Dihydrodiol to AFB1–Dialdehyde, AFB1–Monoalcohols, and AFB1–Dialcohol

The formation of AFB1–dialdehyde occurs non-enzymatically, resulting in two monoalcohols, (Figure 5, reaction #7). However, the dialdehyde itself is reactive and binds to proteins, particularly lysine residues in albumin, and contributes significantly to the acute toxicity of AFB1 [141,142]. There is, however, an aldehyde reductase enzyme that is effective at reducing AFB-dialdehyde to the monoalcohol, and is commonly referred to as “aflatoxin aldehyde reductase”, or AFAR (the *AKR7A1* gene in humans) [141,142]. AFAR reduces AFB-dialdehyde to the C-8, and to a lesser extent, C6a, AFB-monoalcohols. AFAR is also able to further reduce C-8 monoalcohol to AFB-dialcohol [141].

#### Role of Specific Isoforms of Aldo-Ketoreductases in Reducing AFB1–Dialdehyde, by Species

Humans

The human aldo-ketoreductase gene, *AKR7A1*, has been identified as the specific AKR form with activity toward AFB-dialdehyde [143]. Guengerich et al. [141] provided a detailed description of both the non-enzymatic and the enzymatic processes that form and eliminate the reactive AFB1–dialdehyde via AFAR. A second gene coding for a protein with hAFAR-like activity, *AKR7A3*, was cloned and expressed in 1999 [144]. A preliminary comparison of the catalytic properties of the two human AFARs found that hAFAR1 (AKR7a1) was less active than hAFAR2 (AKR7A3) toward AFB1–dialdehyde [141].

b.Non-human primates

No information was identified that addressed specific forms of aldo-ketoreductases in non-human primates that reduce AFB-1-dialdehydre to AFB-alcohols.

c.Rats

A direct comparison of the kinetic characteristics of rat AFAR and human AFAR (AKR7A1) found that the rat form was very similar to the human form. No significant species difference was evident [141].

The rat AFAR1 gene is approximately 8 kb long and comprises seven exons and six introns. It also contains an Antioxidant Response Element (ARE) located in the 5′-regulatory region, and is highly responsive to ARE activators such as ethoxyquin [143]. The substrate specificity of rat AFAR demonstrated that it has broad substrate specificity toward a variety of aldehydes and carbonyls, with highest specific activity toward 9,10-phenanthrenequinone (9,10-PQ). The activity toward 9,10-PQ was ~650 times higher than that of aflatoxin aldehyde [145]. Although the constitutive expression of AFAR in rat liver is low [145], induction of AFAR in vitro has been shown to reduce the acute toxicity of AFB1, and was hypothesized to reduce carcinogenic effects of AFB1, along with the parallel induction of certain GST enzymes [146].

To test the hypothesis that AFAR1 could offer protection against the acute and/or chronic toxicity of AFB1 in vivo in the rat, a liver-specific AKR7A1 transgenic Sprague-Dawley rat was constructed. Two lines, AKR7A1(Tg2) and AKR7A1(Tg5), were found to overexpress AKR7A1 by 18- and 8-fold, respectively. Rates of formation of AFB1 alcohols, as estimated from both in vitro hepatic cytosol assays and in vivo urinary excretion products, were increased in the transgenic lines. However, the substantial over-expression of AKR7A1 in the rat did not provide protection against acute AFB-induced bile duct proliferation (a functional assessment of acute hepatotoxicity by AFB1), nor did it protect against the formation of GST-P positive putative preneoplastic foci as a result of chronic exposure to AFB1 [147,148].

d.Mice

No studies evaluating specific aldo-ketoreductases in mouse were identified, although Ellis and Hayes [145] compared rat AFAR activity toward six other substrates with the mouse aldo-ketoreductase enzymes AR1, AR2, and DD1. The substrate specificity of AFAR was substantially different from any of the three mouse enzymes.

e.Fish/Salmonids

Trout liver cytosol displays slow but detectable aldo-ketoreductase activity toward AFB1 [149], but no information is available on the identity of this AKR or its role in AFB1-induced HCC in this species.

f.Avians (Turkeys, Chickens, Ducks)

Murcia and Diaz [135] reported that hepatic AFAR activity is correlated with acute toxicity of AFB1 in chickens and ducks. These results suggest that AFAR activity is related to resistance to the acute toxic effects of AFB_1_ in chickens and ducks, but not in quail and turkey.

### 4.3. Step #8 Conjugation of AFBO with Glutathione by Glutathione S-Transferases (GSTs)

As discussed above for reaction #1 (Figure 1), the formation of the highly reactive and genotoxic AFBO is an essential step in both acute toxicity and carcinogenesis of AFB1. Although AFBO is highly reactive and unstable in an aqueous environment, it is able to form mutagenic adducts with DNA (discussed in detail later) unless it is either hydrolyzed non-enzymatically, or enzymatically via mEH (Figure 5, reaction #6), or unless it is trapped via reaction with reduced glutathione (GSH; Figure 5, reaction #8). Although some reactive electrophiles can interact with GSH non-enzymatically, little, if any, AFB-GSH conjugate is formed in the absence of GSTs.

Of all the biotransformation pathways characterized for AFB1, the GST-mediated conjugation of AFBO exhibits the largest variability among species, as discussed below. The critical role of GSH in protecting mice from the toxic effects of AFB1 was demonstrated in a comparative study of rats and mice given AFB1. Covalent binding of AFB1 to hepatic DNA in the resistant species, the mouse, was 1.2% of the binding observed in the susceptible species, the rat [150]. The mouse also had 52 times greater in vitro hepatic GST activity toward the AFBO compared to the rat. When GSH was depleted by pretreatment with a combination of buthionine sulfoximine (BSO), an inhibitor of GSH biosynthesis, and diethylmaleate, which depletes existing stores of GSH, AFB-DNA binding in the mouse was increased by 30-fold [150,151].

These and numerous other studies have demonstrated that GST-mediated detoxification of AFBO is a major determinant of species differences in sensitivity to AFB1 hepatotoxicity and hepatocarcinogenesis. Slone et al. [152] evaluated the cytosolic fractions of liver from rats, mice, hamsters, and humans (N = 14) for GST activity toward AFBO, using both synthetic AFBO and microsomally generated AFBO (Table 3).

Activity of cytosolic GSTs from mouse liver were ~50-fold higher than from rat liver. Human liver cytosolic GSTs had little or no measurable activity toward AFBO. A small level of GST activity toward a relatively high concentration (0.4 mM) of synthetic AFBO was detectable in human liver cytosol. Among the 14 different liver samples analyzed, there was a 58-fold variability in AFB-GSH conjugating activity, suggesting that genetic variability may play an important role in determining inter-individual differences in susceptibility to AFB1 toxicity and carcinogenicity [152] (discussed further below).

Kirby et al. [153] also compared cytosolic GST activity toward microsomally generated AFBO in six different species with known or suspected differences in susceptibility to AFB-induced liver tumors. They found that AFB-GST activity in the mouse was 10-fold higher than rat, whereas little or no detectable activity was seen in duck, trout, and human liver cytosolic fractions. The results further demonstrated the large species differences in AFB-GST activity that likely play an important role in determining relative species susceptibility to the carcinogenic effects of AFB1.

#### Role of Specific Isoforms of GSTs in Conjugation of AFBO, by Species

Humans

There are 16 different human genes that code for cytosolic GSTs. Nomenclature has typically classified GST genes into seven classes, using Greek alphabet letters to denote each class: alpha (A), mu (M), omega (O), pi (P), sigma (S), theta (T), and zeta (Z) [154,155]. The alpha class contains five different genes, GSTA1, A2, A3, A4, and A5. As shown in Table 2, human cytosolic GST activity toward AFBO is extremely low—below the limit of detection in assays using microsomally generated AFBO [152].

Buetler et al. [49] evaluated bacterially cDNA-expressed and purified human GSTA1-1 and GSTA2-2 activity toward AFBO generated from mouse liver microsomes and found a little activity, which was detectable, but below the limit of quantitation. Raney et al. [52] measured human GST activity toward synthetic exo- and endo-AFBO with purified human alpha-class GSTs A1-1 and A2-2 and also found measurable, but very low, activity compared to rat alpha-class GSTA1-1. They also measured AFBO-conjugating activity for purified human GSTM1a-1a and GSTP1-1. GSTM1a-1a had measurable activity toward exo-AFBO, whereas GSTP1 had no detectable activity. Johnson et al. [156] performed a comparative kinetic analysis of cDNA-expressed and purified human GSTs A1-1, A2-2, M1-1, P1-1, and T-1-1, as well as rat GSTA5-5. hGSTM1-1 had activity (catalytic efficiency) about 20-fold greater than any of the other human GSTs, but it was ~2000 times lower than the high-activity rGSTA5-5. They did detect a small amount of exo-AFBO activity in recombinant hGSTA1-1 that was ~6-fold lower than GSTM1-1.

The identification of human GSTM1-1 as the only human liver GST with significant activity toward exo-AFBO raises interesting questions regarding potential genetic variability in susceptibility to AFB1, since the human GSTM1 gene is highly polymorphic, with ~50% of the human population homozygous-null for the GSTM1 gene. Several studies have examined this hypothesis. Gross-Steinmeyer et al. [157] measured the extent of AFB-DNA adducts formed in primary human hepatocyte cultures incubated with 0.4 µM ^3^H-AFB1 for 6 h (Figure 7).

There was a 3-fold reduction in AFB-DNA adducts in hepatocytes from GSTM+ livers, compared to livers that lacked the GSTM1 gene (GSTM1-null) [157].

These and other studies [158,159] provide some evidence to indicate that individuals who are GSTM1-null may be at increased risk of AFB1-induced HCC. However, Stewart et al. [159] evaluated human lung samples for the ability to both activate AFB1 to AFBO and to conjugate AFBO with cytosolic GSTs. At a relatively high in vitro concentration of AFB1 (100 µM), AFB-GSH conjugate was seen in all 10 lung samples, but with very large interindividual variability. They genotyped the samples and found no evidence that GSTM1-null lung samples produced less AFB-GSH. They noted that other GSTM forms (GSTM3, GSTM4) are present in human lung tissue, even in GSTM1-null samples, but there is no information on the relative potential of GSTM3 to conjugate AFBO.

b.Non-human primates

Wang et al. [160] purified several hepatic GSTs from the non-human primate, *Macaca fasicularis* (mfaGST) and assessed their ability to conjugate AFBO. Two GSTs exhibited significant AFBO-conjugating activity and were identified by Western blot analysis as belonging to the mu class of GSTs. Similar to those of humans, purified Macaca liver alpha-class GSTs had no detectable AFBO activity. Interestingly, the two Macaca mu-class GSTs differed in their enantioselectivity: one form had higher activity toward the exo-AFBO, and the other had higher activity toward the endo-AFBO. Two cDNA clones of mu-class Macaca GSTs were generated and expressed; one had high activity toward AFBO, and the other had no measurable activity. The high AFBO-activity form shared 86% and 94% sequence homology with hGSTM1a and hGSTM2, respectively. GSTM2 is not expressed significantly in human liver, so does not likely contribute to protection from AFB1. However, the enantioselectivity of the cDNA-expressed mfaGSTM2 was different from the purified mu-class GST, and other differences suggested that the GST protein from the cloned cDNA was not identical to the purified mu-class GST with high AFBO activity. Nevertheless, this paper demonstrated that, like humans, the non-human primate hepatic GST with substantial AFBO activity belongs to the mu-class of GSTs. A subsequent paper cloned and expressed the *Macaca fasicularis* alpha-class GST with homology to human GSTA1 and was shown not to possess any measurable activity toward AFBO [161].

c.Rats

The AFB-GSH conjugate is one of the major metabolites of AFB1 formed in vivo in the rat and is excreted in bile [111,162]. Several studies have evaluated the specific activity of rat GSTs toward AFBO [52,154,163,164,165,166]. The major GST expressed constitutively in rat liver is GSTA3-3 [167], but it exhibits less than 0.1% activity toward AFBO compared to mGstA3-3, even though they have similar activity toward the generic GST substrate, 1-chloro-2,4-dinitrobenzene (CDNB) [165,166].

However, another rat GST gene, *GSTA5*, codes for a protein (rGSTA5-5) with high specific activity toward AFBO and is the orthologous rat gene to the mouse *GstA3* gene, which codes for a constitutively expressed protein with extraordinarily high AFBO activity (see discussion below on mouse GSTs). In rats, protection from aflatoxin-induced carcinogenicity may be provided by the induction of rGSTA5 (previously termed rYc2), with high AFBO-conjugation activity [168]. The rGSTA5 subunit shares 91% sequence identity with mGSTA3 and is inducible by ethoxyquin and other antioxidants [169].

Induction of the *rGSTA5* gene with activating ligands for the antioxidant response element pathway (KEAP1/NRF2) greatly reduces AFB-DNA damage in rats, rendering them resistant to the development of preneoplastic lesions and liver tumors [11,170,171].

d.Mice

As discussed above, one specific form of GST in the mouse, mGstA3-3, has been demonstrated to have very high activity towards AFBO and affords extensive protection against AFB1 carcinogenesis in the mouse [172,173]. The mGstA3 gene has been cloned and sequenced [164,174]. The mouse gene is located on chromosome 1, region A3-4, and is encoded by seven exons, spanning approximately 24.6 kb of genomic DNA [175]. A *mGstA3* knockout (KO) mouse has been developed and shows very high sensitivity to AFB-DNA adduct formation, compared to the wild-type, with DNA adduct formation in the KO mouse 112 times greater than in the wild-type (Figure 8) [173].

Amino acid sequence comparison of the high-AFBO-activity mouse GstA3-3 protein with the low-AFBO-activity rat GSTA3-3 protein found only 15 amino acids that are different. Construction of a chimeric rGSTA3 (high AFBO activity) and rGSTA3 (very low AFBO activity) demonstrated that much of the high activity domain of the protein was in the last two-thirds of the protein sequence [165]. Site-directed mutagenesis of the low-activity rGSTA3 gene to different amino acids that were conserved in both the high-AFBO-activity mGstA3-3 and rGSTA5-5 protein identified six amino acids that combine to confer a 2000-fold increase in catalytic activity toward AFBO, approaching that seen in rGSTA5-5, but still 6-fold less than the mGstA3-3 protein (Table 4) [166].

One amino acid change, E208D, increased AFBO activity about 100-fold, but additional amino acid changes conferred relatively modest increases, although showing synergistic interactions (Table 3). Surprisingly, adding two additional amino acid changes in the low-activity gene to the high-AFBO sequence (V106Y and Y111H) decreased AFBO-conjugating activity from 40.0 to 2.8 pmol/mg/min [166]. There was no consistent effect on CDNB activity, demonstrating substrate selectivity toward AFBO [166].

Thus, the ~2000-fold differences in AFBO-conjugating activity between the constitutively expressed *mGstA3* gene and the major constitutively expressed alpha-class GST in rats, *rGSTA3*, largely if not completely explains the dramatic difference in sensitivity between mice and rats to the toxic and carcinogenic effects of AFB1.

e.Fish/Salmonids

Conjugation of AFB1-8,9-epoxide with GSH by hepatic GSTs is a minor pathway for AFB1 metabolism in vitro and in vivo [68] and this lack of efficient detoxication is a major factor in the high sensitivity of trout to AFB1-induced HCC.

f.Avians (Turkeys, Chickens, Ducks)

Early studies conducted to determine the biochemical basis for the extreme sensitivity of domesticated turkeys to AFB1 revealed a lack of hepatic cytosolic GST-mediated conjugation activity of exo-AFBO, even in metabolic studies using [^3^H]-AFB1 as a substrate to enhance sensitivity [73,78] (Figure 9).

Because of the well-established centrality of *GSTs* in species susceptibility, Kim et al. cloned and expressed and functionally characterized six *GSTAs* from the livers of domesticated and wild turkeys [176]. *GSTAs* contained alpha-class conserved domains, signature motifs, and numerous SNPs in their coding regions: *GSTA1.1*, *GSTA1.2*, *GSTA1.3*, *GSTA2*, *GSTA3*, and *GSTA4*. When expressed in *E. coli*, all six *tGSTA* gene products were active toward GST prototype substrates, confirming the catalytic specificity of the GSTA constructs. Activities of cDNA-expressed recombinant GSTAs, as well as their hepatic cytosolic forms, exhibited catalytic activity toward prototype substrates CDNB, DCNB, ECA, and CHP. The highest activity toward CDNB was observed for tGSTA1.2 cloned from domesticated turkey. The GSTA3 allele cloned and expressed from Royal Palm turkeys had substantially higher ECA and CHP activity than all other recombinant GSTs. As we have previously observed in GSTA1.1 from domestic turkey [177], all GSTA1.1 isoforms lack a signature motif (PVxEKVLKxHGxxxL; residues 134–148) in the C-terminal domain. However, this did not appear to discernibly affect the catalytic activity of these recombinant, expressed isoforms. Indeed, the specific enzyme activities of GSTA1.1 were well within the range of that seen from other isoforms.

Importantly, recombinant GSTAs possessed detoxification activity toward in situ-generated exo-AFBO-reactive intermediate. Recombinant GSTA1.1 and A1.3 from Eastern and Rio Grande wild turkeys appeared to have the highest AFBO-trapping activity. Surprisingly, unlike their hepatic forms, recombinant *E. coli*-expressed *GSTAs* from domesticated turkeys possessed AFBO-trapping activity, on par with that of wild turkeys [178] (Table 5). The observation that recombinant tGSTAs from domesticated turkeys detoxify AFBO, whereas the tGSTs expressed in vivo in the liver do not, implies that the tGST enzymes with AFBO activity are not constitutively expressed and thus must be down-regulated, silenced, or otherwise modified by one or more mechanism(s), possibly epigenetic in nature.

Loss of protective GST alleles in domesticated turkeys through genetic selection for commercially desirable traits is a plausible explanation for their extreme sensitivity compared to wild birds. By comparison, liver cytosol from BHA-induced mice, which expresses high amounts of mGSTA3, the “gold standard” of AFBO-conjugating activity, possessed AFBO-conjugating activity that was 41-, 44-, and 26-fold greater than that in livers from Eastern, Royal Palm, and Rio Grande wild turkeys, respectively. The rate of AFBO conjugation measured in mouse liver [178] was close to that published for similarly BHA-induced Swiss Webster mice [105].

Because human and non-human primate mu-class GSTs have been shown to have catalytic activity toward exo-AFBO [33,156,160], studies were conducted to determine whether GSTM genes in turkeys might also have a role in AFBO detoxification. Livers from domesticated and wild turkeys express two hepatic *mu*-class tGSTs (tGSTMs) which we amplified, cloned, expressed, and functionally characterized [177,178,179]. These were *tGSTM3* and *tGSTM4* from domesticated turkeys, and a *GSTM4* variant (*ewGSTM4*) from Eastern wild turkeys [179]. The predicted molecular masses of tGSTM3 and two tGSTM4 variants were 25.6 and 25.8 kDa, respectively. Multiple sequence comparisons revealed four GSTM motifs and the mu-loop in both proteins. tGSTM4 has 89% amino acid sequence identity to chicken GSTM2, while tGSTM3 has 73% sequence identity to human GSTM3 (hGSTM3). Specific activities of *E. coli*-expressed tGSTM3 toward CDNB and peroxidase activity toward cumene hydroperoxide were five-fold greater than those of tGSTM4, while tGSTM4 possessed more than three-fold greater activity toward DCNB. The two enzymes displayed equal activity toward ECA. However, unlike the turkey GSTAs, none of the recombinant GSTMs had AFBO detoxification capability. In total, these data indicate that although turkey hepatic GSTMs may contribute to detoxification, they probably play no role in detoxification of AFBO in the liver [179].

### 4.4. Step #9 Formation of the AFB-NAC Conjugate from AFB-GSH

Once formed, the AFBO-GSH conjugate is subject to transmembrane transport and subsequent enzymatic modifications to the mercapturic acid (Figure 5, reaction #9) [180]. Following conjugation in the hepatocyte, AFB-GSH is transported out of the hepatocyte at the canalicular membrane into the bile. Although no studies were identified that evaluated which of several possible canalicular transporters move AFB-GSH into bile, hydrophobic glutathione conjugates such as AFB-GSH are generally excellent substrates for the multidrug-resistance-associated protein-2 (MRP2) [181,182]. During transport into bile, biliary and/or canalicular γ-glutamyl transferase (γ-GT) removes the y-glutamyl moiety from the AFB-GSH to form a cysteinylglycine S-AFB conjugate (AAFB-Cys-Gly). Moss et al. [183] found that the ratio of AFB-GSH to AFB-Cys-Gly in the bile of rats administered AFB1 intravenously was 5.5:1. They found that inhibition of γ-GT in the rat prior to administration of AFB1 completely eliminated the presence of AFB-Cys-Gly in the bile.

Dipeptidase (DP) present in bile and intestinal tissue then removes the glycine moiety, resulting in a cysteine S-conjugate, which is then absorbed and transported back into the liver [181,182]. In the cytosol, *N*-acetyl transferases create AFB-N-Acetylcysteine (NAC; aka AFB-mercapturic acid), which is more polar and more water soluble than AFB-GSH. AFB-NAC is likely to be non-toxic and excreted from the body through bile or urine.

There is little specific information known about species differences in the formation and excretion of AFB-GSH and its metabolites (AFB-Cys-Gly and AFB-NAC). Since the products of AFB-GSH catabolism are likely of little to no toxicological consequence, we will not discuss this pathway by specific species.

In 1997 the Groopman lab [184] reported on a sensitive LC-ESI/MS method to measure both endo- and exo-AFB-NAC in the urine, with potential use as a biomarker of exposure. After administration of 40 μg of AFB1 for 5 days/week for 8 weeks to rats, urine was collected and analyzed for both endo- and exo-AFB-NAC. About 1% of the administered dose was recovered in urine as AFB-NAC, of which ~90% was the ex-AFB-NAC. Using this assay, Kensler et al. [185] reported the urinary excretion of AFB-NAC was >200-fold higher in wild-type mice compared with single GSTA3 knockout mice, further highlighting the importance of this GST as a key determinant in the resistance of mice to AFB1. An immunoassay for AFB-NAC was also developed [186] and proposed for use as a biomarker of AFB1 exposure. AFB-NAC has been used as a biomarker of AFB1 exposure in several epidemiology studies [126,187]. Wang et al. [187] reported a 2.6-fold increase (*p* = 0.017) in the median rate of AFB-NAC excretion in participants randomized to receive the drug oltipraz daily for 4 weeks compared to placebo. Oltipraz is an effective inhibitor of AFB1 hepatocarcinogenesis in rats, an action likely mediated by induction of GSTs in this species [188]. Perhaps some isoform of human GST can catalyze the conjugation of AFBO under conditions of low ambient AFB1 exposures present in this Chinese study population at the time of the trial. (See also Kensler and Eaton [189] in this *Special Issue* for a discussion of urinary biomarkers of AFB1 exposure.)

Beyond these studies, there is relatively little species-specific information on this pathway.

### 4.5. Step #10 Glucuronide and Sulfate Conjugation of Hydroxylated AFB1 Metabolites

As noted previously, there is substantial evidence that the glucuronide conjugate of AFP1 is a major biliary metabolite of AFB1 in rats [111,190,191,192]. AFP1 is a minor oxidative metabolite in humans, and there is little information on which UGT enzyme(s) in humans or other species may form the glucuronide conjugate of AFP1. Several early studies examining the biotransformation of AFB1 reported that the hydroxylated products of AFB1 form glucuronide and sulfate conjugates (Figure 5, reaction #10) [193,194,195,196,197].

Several studies have also shown that direct conjugation of AFB1, or more likely oxidative metabolites of AFB1, with glucuronic acid and sulfate, occurred both in vivo and in hepatocyte cultures [191,192,197,198,199]. Dalezeos and Wogan [194] reported that AFP1 and its glucuronide and sulfate conjugates were found in the urine of Rhesus monkeys given AFB1. Approximately 50% and 10% were AFP1–glucuronide and AFP1–sulfate, respectively. The balance was unconjugated AFP1.

Wei et al. [192] conducted an extensive analysis to determine the presence of chloroform-extractable and water-soluble AFB1 metabolites in the urine of Rhesus monkey, rat, and mouse given ^3^H-AFB1. Conjugates of AFM1 were the principal AFB1 metabolites in the urine, accounting for about 20% of total urinary radioactivity, but the hydrolysis procedure used a combination of glucuronidase and sulfatase, so it was not possible to distinguish the contribution of the two. AFP1-conjugate(s) accounted for ~2%, and no AFQ1 conjugate was identified in the urine of monkeys. In rat urine, there was somewhat more AFP1-conjugate(s) than AFM1-conjugate(s), but the fraction of total conjugates was substantially less than seen in either the monkey or mouse urine. Mouse urine contained substantially more water-soluble metabolites, with the distribution of conjugates between AFM1, AFP1, and AFQ1 being similar. There was also a significant quantity of unidentified polar metabolite(s), which may have been the AFB-GSH and AFB-NAC metabolites.

Interestingly, there is some evidence from early studies that AFB1 itself may be subject to glucuronidation and/or sulfation following enol-keto tautomerism of the cyclopentane keto group to a hydroxyl group [192,197]. These authors have suggested that this may contribute to AFB1 toxicity at sites other than the liver, following enzymatic or acid hydrolysis of the conjugate to release AFB1 in other tissues.

Overall, there is relatively little evidence that species differences in glucuronidation and/or sulfation of phase I products of AFB1 metabolism contribute significantly to differences in AFB1 carcinogenicity or other toxic effects.

## 5. Summary

It is now 65 years on from the discovery of AFB1 as a contaminant of groundnut meal, and identification of its toxic potential. A search of PubMed retrieved 16,741 papers related to aflatoxins between 1960 and the present (Figure 10).

It is evident by the timeline that the scientific world has not lost interest in the public health, veterinary, and agricultural significance of aflatoxins. In the last 5 years, there have been ~750 papers per year, whereas in years prior to 2010, the average number of papers was less than 300 per year. The influence of Professor John D. Groopman’s laboratory since he began his training in the laboratory of Dr. Gerald N. Wogan in the late 1970s is illustrated by his publication and citation record (Figure 11). A Web of Science search for Dr. Groopman’s publications between 1979 and the present identified 219 publications, with over 7800 papers citing his work (nearly 13,750 total citations). From 1979, when Dr. Groopman published his first paper, to the present, there have been approximately 14,700 publications listed on PubMed on some aspect of aflatoxins.

It is striking to consider that over 50% of these articles (~7800 papers) cited one or more of Dr. Groopman’s publications—a tribute to the tremendous impact his work has had on this field.

The importance of species differences in susceptibility to the toxic and carcinogenic effects were recognized almost immediately following early investigations into the effects of aflatoxins on turkeys, ducklings, trout, rats, and mice. It was also evident early on that the biotransformation (referred to as metabolism in most of the older literature) was likely to be the key determinant of species differences in susceptibility to AFB1. Some 60+ years later, the scientific community has largely solved this mystery. We provide here a quick summary of what we now know, by species:Humans and non-human primates

The hepatotoxic and carcinogenic effects, and likely all or nearly all of the other toxic effects, of AFB1 require oxidative biotransformation to AFBO. The exo-conformer of AFBO binds to DNA, causing mutations and cytotoxicity that represent the key molecular initiating event in AFB1 carcinogenesis. Both the endo and the exo conformers form adducts with proteins, which contribute to its hepatotoxicity and likely other toxic effects observed in human populations exposed to AFB1 via the diet. Urinary and blood biomarkers of exposure and effect have been developed by Drs. Groopman and Kensler’s laboratories, along with many others across the world [189].

In humans, the formation of both endo- and exo-AFBO is mediated largely, if not exclusively, by two human CYP enzymes. hCYP1A2 forms roughly equivalent amounts of exo- and endo-AFBO, whereas hYP3A4 forms mostly exo-AFBO. However, there are important kinetic differences in the affinity of AFB1 to the active sites of these two CYPs. hCYP1A2 is expressed only in the liver, and in lower amounts than hCYP3A4, and thus has a considerably lower Vmax. Although there is more CYP3A4 enzyme available in the liver (and other tissues) to oxidize AFB1 to AFBO, the enzyme follows Hill kinetics, rather than classic Michaelis–Menten kinetics, and thus exhibits “self-activation” kinetics highly dependent on substrate concentration. At low concentrations in the liver (e.g., less than 0.1 µM), the enzyme is largely inactive toward activation of AFBO. But at concentrations above ~10 µM, it dominates exo-AFBO formation. In contrast, hCYP1A2 follows classic Michaelis–Menten kinetics and exhibits a high affinity (low Km) for AFB1, such that, at the low substrate concentrations that would be seen in human liver following dietary exposure, it dominates AFBO formation.

In addition to AFBO, human CYPs are responsible for the formation of the less toxic AFB1 metabolites, AFM1, AFP1, and AFQ1. AFM1 is formed by both hCYP1A1 and 1A2, but hCYP1A1 is generally expressed at very low levels in human liver, at least in the absence of exposure to AhR inducers, such as numerous polyaromatic hydrocarbon. Thus, it is likely that in most people, AFM1 is formed principally by hCYP1A2. AFM1 is capable of activation to the epoxide, likely via secondary oxidation by hCYP1A2, and thus has some carcinogenic effects, but is about 10-fold less potent than AFB1. AFM1 is the primary AFB1 metabolite found in urine, and has been used by Dr. Groopman and others as a biomarker of exposure in some studies.

AFP1 results from oxidative demethylation in the liver, and appears to be a relatively minor oxidative metabolite in humans, although a major metabolite in rats and mice (see below). It is uncertain which human CYP(s) form AFP1.

AFQ1 is a major oxidative metabolite of AFB1 and is formed largely, if not exclusively, by hCYP3A enzymes. hCYP3A4 is the major form expressed in human liver, and forms approximately 10 times as much AFQ1 as exo-AFBO. In that capacity, hCYP3A4 may serve predominantly as a detoxification pathway. hCYP3A5 is a polymorphic enzyme that is expressed only in a relatively small portion of the population, but does exhibit similar activity toward AFB1 as hCYP3A4. hCYP3A7 is a fetal form and thus found only in fetal tissues, but it, too, seems to have similar activity toward AFB1 as hCY3A4.

There is one reduction pathway involved in AFB1 biotransformation that may occur in humans—reduction of the cyclopentanone moiety to a hydroxyl group, referred to as aflatoxicol (AFL). AFL is capable of undergoing oxidation back to AFB1, so this pathway could potentially serve as a “reservoir” for AFB1, allowing exposure to tissue other than the liver to AFB1 if oxidized back to AFB1. There is relatively little information on the aldo-ketoreductase(s) that is/are responsible for this pathway in humans. It appears that this is a minor pathway, and probably not of great importance in the overall disposition of AFB1 in humans. There is also little information on relative species differences.

AFBO, once formed, is subject to hydrolysis to the AFB1–dihydrodiol. This reaction likely occurs both non-enzymatically through the addition of water across the epoxide bond, but there is some evidence that microsomal Epoxide Hydrolase (mEH) can also mediate this reaction. There are limited data that suggest that human mEH may offer protection against AFBO-DNA damage and mutagenicity.

AFBO is also subject to conjugation with reduced glutathione (GSH). Although exo-AFBO is unstable in an aqueous environment and highly reactive, it does not appear to react non-enzymatically with GSH. The conjugation requires specific glutathione S-transferase (GST) enzymes to detoxify both endo- and exo-AFBO. In humans, all alpha-class GSTs seem to lack catalytic activity toward AFBO. However, hGSTM1-1, a mu-class GST, does have low but measurable activity toward AFBO. This enzyme is polymorphic in the human population, with ~50% of the overall population having a “homozygous null” genotype, and thus no enzyme is produced. There are some data showing that individuals who are GSTM1-null may be at increased susceptibility to AFB1.

The formation of AFB1–dihydrodiol from either enzymatic or non-enzymatic reactions plays an important role in AFB1 toxicity because aldo-keto tautomerization results in the formation of AFB1–dialdehyde, which is reactive and can bind to proteins.

However, two human aldo-ketoreductases, called ”aflatoxin aldehyde reductase” or AFAR1 and AFAR2 (from the human genes *AKR7A1* and *AKR7A3,* respectively) can reduce the reactive dialdehyde to AFB1–dialcohol and AFB1–monoalcohol (two forms). Because the alcohols do not bind to proteins, this is considered a detoxification pathway. In the absence of significant GST activity in humans, this pathway may play a significant role in detoxification of AFBO by reducing protein binding.

Relatively little is known about glucuronide and sulfate conjugates of the various hydroxylated AFB1 metabolites, other than some glucuronide conjugates of AFP1 have been identified at low levels in urine. However, with the possible exception of AFM1, the hydroxylated AFB1 metabolites seem not to contribute significantly to AFB1 carcinogenesis and toxicity, and thus potential interindividual and species differences in glucuronidation of aflatoxin metabolites are not likely to contribute significantly to species differences in AFB1 toxicity.

b.Rats

Overall rat hepatic CYP enzymes (microsomal fraction) have about twice the AFBO-forming activity as human liver microsomes. It is less clear which specific CYPs in rats are responsible for AFBO formation, although the rat orthologue of hCYP3A4, rCYP3A2, may be responsible. In contrast to hCYP1A2, rCYP1A2 appears to have little activity toward AFBO formation.

In contrast to humans, rats form relatively large amounts of AFP1, which as the glucuronide conjugate is a major biliary metabolite. At low AFB1 concentrations, AFQ1 is a very minor metabolite in rat hepatocytes, but at higher substrate concentration it accounts for about 20% of total metabolites, suggesting that ratCYP3A enzyme has similar kinetic characteristics to human.

The constitutively expressed GSTs in rat liver (*rGSTA1* and *rGSTA2*) have relatively little ability to detoxify AFBO. However, rGSTA5-5, which is inducible by activating ligands of the KEAP1/NRF2 antioxidant response pathway, has very high activity toward AFBO. Induction of rGSTA5 results in complete inhibition of the carcinogenic effects of aflatoxin. This remarkable protective effect has served as a primary basis for numerous chemoprevention clinical trials.

c.Mice

Mouse liver microsomes have high CYP activity and form AFBO at about twice the rate of rats, and about 5 times that of human liver microsomes. Mouse microsomes form relatively large amounts of AFM1 and AFP1 as well. It is not clear which mouse Cyp enzymes form AFBO, but the limited data available suggest that mCYP3a is likely more important than mCyp1A2.

Although adult mice activate AFB1 to AFBO relatively more rapidly than human or rat microsomes, they are nearly completely resistant to the carcinogenic effects of AFB1. It is now widely recognized that constitutively expressed mGstA3-3 has high activity toward AFBO—at least 10,000-fold higher than human GSTM1-1. mGstA3-3 activity toward AFBO is about 6 times greater than the inducible, highly active form of rats, rGSTA5-5. Studies with m*Gsta3* knockout mice, coupled with studies showing induction of rGSTA5-5, indicate rats, like mice, are resistant to AFB-DNA damage and mutagenesis, confirming that the expression of these two GSTs can account for the gradient in species differences in sensitivity between rodents and then compared to primates.

d.Fish/salmonids

Metabolism of AFB1 plays the key role in determination of species sensitivity to AFB1-dependent HCC. Trout constitutively express a CYP (2K1) with high activity toward production of the exo-epoxide and relatively lower rates of inactivation of the epoxide by conjugation with GSH or production of other phase 1 or phase 2 metabolites. Alteration of this metabolic profile by, for example, BNF or PCBs, induces a CYP1A1-dependent production of AFM1 and markedly reduces AFB1-dependent HCC.

Avians (Turkeys, Chickens, Ducks)

Domesticated turkeys represent an example of an extremely sensitive animal model, owing to, at least in part, to a combination of efficient P4501A5-mediated bioactivation, coupled with deficient GST-mediated detoxification of the AFBO so generated [85]. While domestic turkeys possess constitutive GSTs, none are able to detoxify AFB_1_ in vivo or in vitro [73,78,79]. Conversely, wild turkeys, which are relatively AFB_1_-resistant compared to their domesticated counterparts [76], possess functional, AFB1-detoxifying hepatic GSTs [178]. As in the case of mammalian species, GST-mediated AFB1 detoxification is the rate-limiting determinant in the sensitivity of various avian species, regardless of the extent of P450-mediated bioactivation [82].

## 6. Conclusions

To date, appreciation of the well-established species differences in susceptibility to the carcinogenic effects of AFB1 has not altered most risk assessments or regulatory limits on AFB1. Perhaps fortuitously, many relied upon rat liver tumor studies for quantitative assessment for human cancer risk. For example, a recent European Food Safety Authority (EFSA) evaluation of AFB1 carcinogenicity used a BMDL10 of 0.4 µg/kg bw per day for the induction of HCC by AFB1 in male rats as a reference point [200]. However, a Margin of Exposure (MOE) approach, rather than a linear extrapolation of rat liver tumor data, was used, with a 10,000-fold uncertainty factor to adjust for species differences and other uncertainties. They noted that, for more highly exposed populations, MOEs were frequently less than 10,000.

When comparing relative rates of CYP-mediated activation to exo-AFBO, rat and human liver appear similar. However, rats have significant GST-mediated detoxification, and human liver has remarkably little, suggesting that additional uncertainty factors may be warranted when using rat liver tumor data. Indeed, differences in expression and catalytic function of a variety of biotransformation enzymes can explain in large part, if not exclusively, the well-known differences in susceptibility to hepatotoxicity and carcinogenicity of AFB1. Based on the relatively very low level of human GSTs toward AFBO, it would appear that humans should be highly susceptible to AFB1 toxicity. However, studies in isolated perfused liver and human hepatocytes in culture suggest that the hepatic DNA damage from AFBO in humans is more similar to that in rats than would be expected based solely on hepatic AFBO-GST activity. These and other studies suggest that other pathways, perhaps including mEH, play a relatively more important role in protecting human liver cells from AFBO-DNA damage. There is evidence that human mEH may play a significant role in detoxification of AFBO, especially in the presence of limited AFBO-GST activity. It is also possible that human DNA repair processes are more efficient, adding additional protection.

It is notable that there are sex differences in the susceptibility of rats to aflatoxin hepatocarcinogenesis, with males more susceptible than females. In human populations with known exposures to aflatoxins, there are typically 2–3-fold increased rates of liver cancer incidence in males. However, this outcome is not universal, as in regions of Central America no difference by sex is observed. To date, there is little evidence to suggest that differences in aflatoxin biotransformation are likely to account for increased sensitivity of males to carcinogenesis, be it rats or humans.

Finally, there are other well-established interactive risk factors, such as hepatitis B and C virus infections in humans, that greatly potentiate the hepatocarcinogenic activity of AFB1. Conversely, there is a growing body of literature implicating dietary antioxidants as offering some protection against AFBO genotoxicity by either inhibiting CYP-mediated activation or inducing potential protective pathways. Genetic differences in AFB1 biotransformation pathways (such as the GSTM1 genotype) further complicate the quantitative extrapolation of laboratory animal data to human liver cancer risk. Yet, despite all of these uncertainties, it appears that the rat has been a useful surrogate for risk assessment of aflatoxin carcinogenicity in humans given the dominant role of overall metabolism as the determinant of susceptibility with generally similar activation activity. Although GST-mediated detoxification is much lower in humans than in rats, it appears that alternative detoxification pathways, perhaps including human mEH, compensate for the relative lack of GST activity, giving rise to a relatively similar overall activation: detoxification profile between humans and rats. The remarkable sensitivities of turkeys and rainbow trout provide further demonstration that biotransformation pathways are the major determinants of species susceptibility to AFB1 hepatoxicity and carcinogenesis.

## Figures and Tables

**Figure 1 toxins-17-00030-f001:**
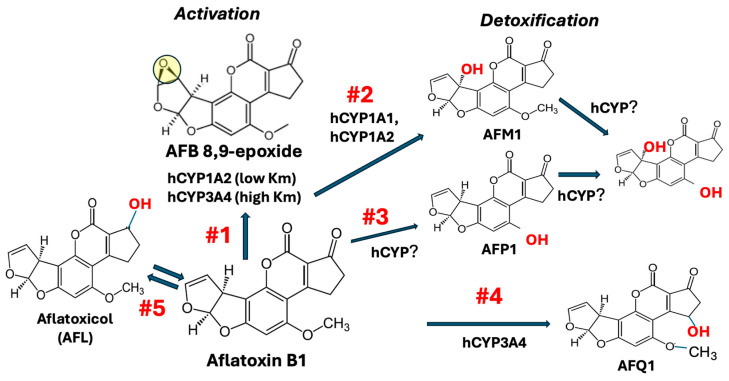
Basic steps in the oxidation of AFB1 to various metabolites. The human enzymes, where known, catalyzing these oxidations are listed. Each oxidation step shown in Figure 1 is discussed in detail below, with a focus on understanding important species differences in each oxidation step, as well as the specific enzyme isoforms that contribute to each reaction.

**Figure 2 toxins-17-00030-f002:**
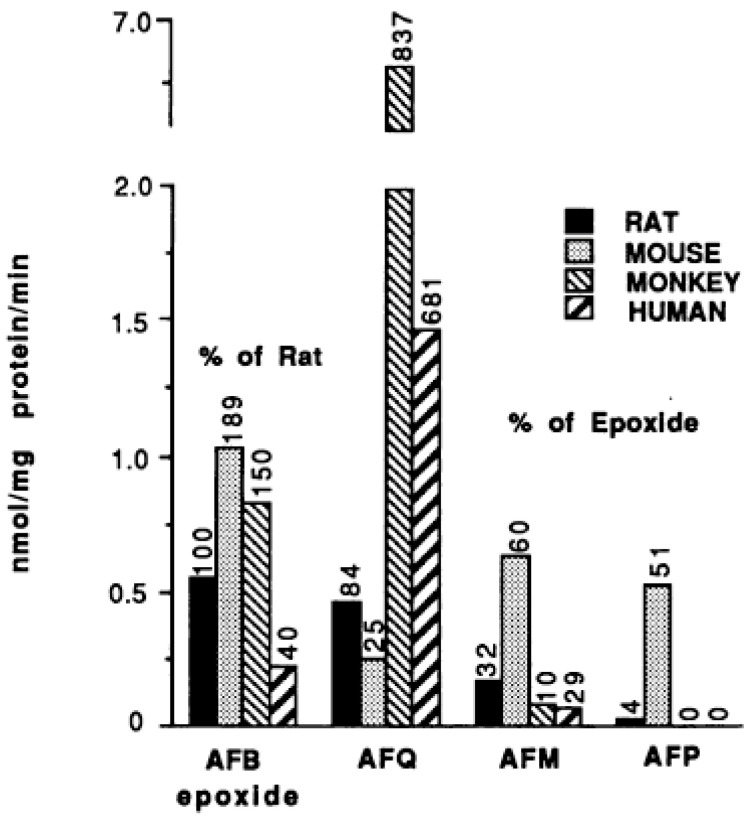
Hepatic microsomal oxidation of AFB1 to various oxidative metabolites in different species. The initial rates of formation (Vo) were determined in hepatic microsomes from rat, mouse, monkey, and human microsomes under identical experimental conditions. AFBO was determined by trapping as the GSH conjugate using BHA-induced mouse liver cytosol, which contains a high level of mGSTA3-3. Each metabolite was separated and quantitated by HPLC. Rates of AFBO formation as a percentage of that observed with rat liver microsomes are also shown. The rates of formation of AFQ1, AFM1, and AFP1 were calculated as a percentage of the rate of epoxidation observed for the respective species; these values are shown above each column. (From Ramsdell and Eaton [16]). Reprinted under AACR copyright permissions to authors.

**Figure 3 toxins-17-00030-f003:**
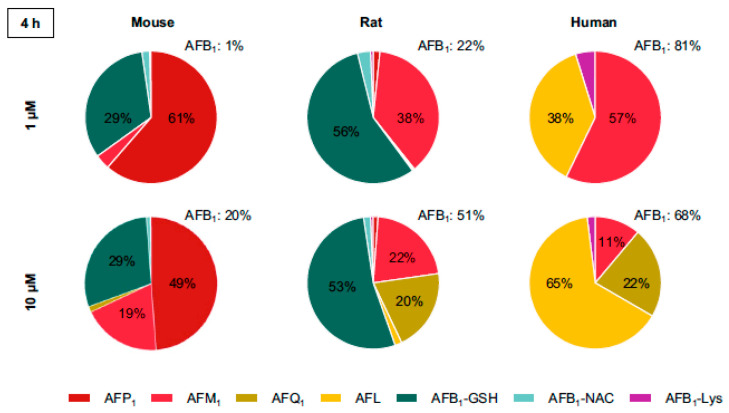
AFB1 metabolite distribution at 1 μM and 10 μM in mouse, rat, and human hepatocytes. Isolated hepatocytes from each species were incubated for 4 h in cell culture medium. Metabolites were identified by HPLC-MS/MS. (From: Gerdemann et al. [6]; figure is reprinted under Creative Commons Attribution 4.0 International License).

**Figure 4 toxins-17-00030-f004:**
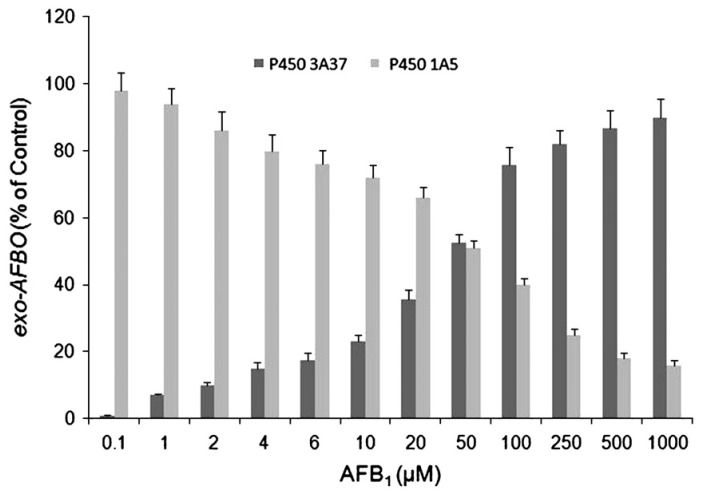
Immuno-inhibition experiments using anti-peptide antiserum against turkey P450s 1A5 and 3A37 demonstrating the relative contribution of P450 1A5 and 3A37 toward AFB1 epoxidation in turkey liver microsomes. Inhibitory effects of anti-P450 1A5 and 3A37 immune serum (5 μg/mL/nmol P450). Initial rates of exo-AFBO formation in the presence of antiserum were calculated as percentage control (treatment with pre-immune serum only). Mean ± SD. (N = 3). From: Rawl and Coulombe, [85]. Reprinted under Open Access Creative Commons Attribution.

**Figure 5 toxins-17-00030-f005:**
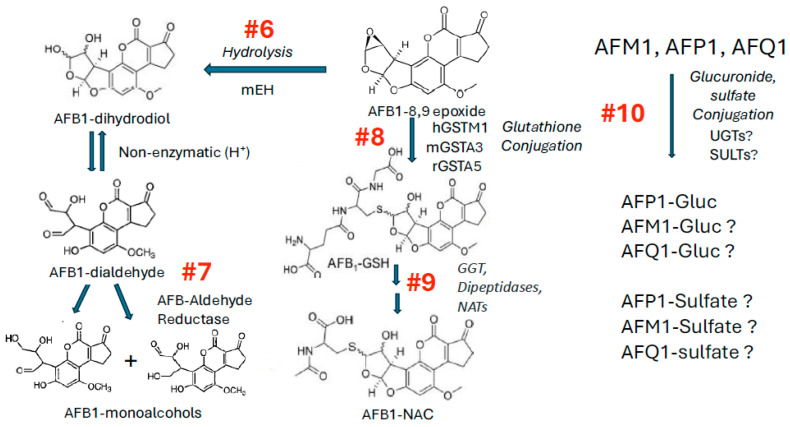
Phase II hydrolysis and conjugation reactions of phase I oxidation products of AFB1 biotransformation.

**Figure 6 toxins-17-00030-f006:**
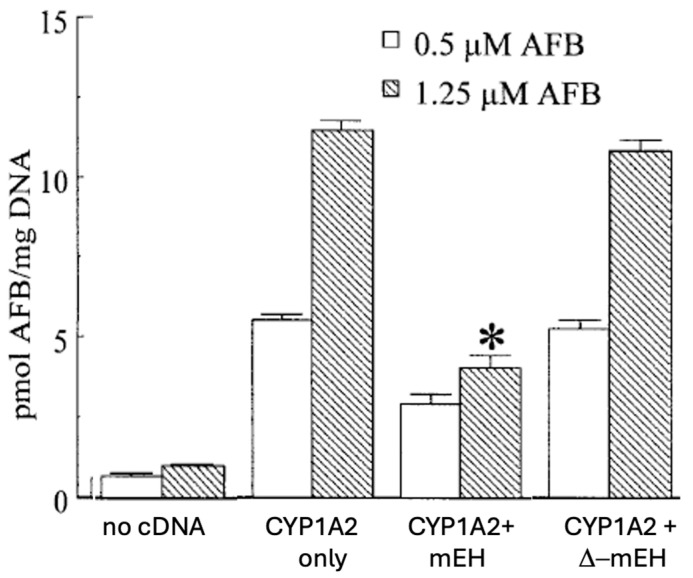
Effects of co-expression of human mEH on AFB-DNA adducts in yeast also co-expressing hCYP1A2 to activate AFB1 to AFBO. Two concentrations of AFB1 were used to expose yeast cells containing human CYP1A2 and mEH cDNAs (adapted from Kelly et al. [138]. * Co-expression of mEH blocked DNA adduction with significant effect (*p* < 0.05) at 1.25 mM AFB. Data are mean 6 SEM from samples analyzed in triplicate. (Figure available under Creative Commons Attribution 4.0 International license).

**Figure 7 toxins-17-00030-f007:**
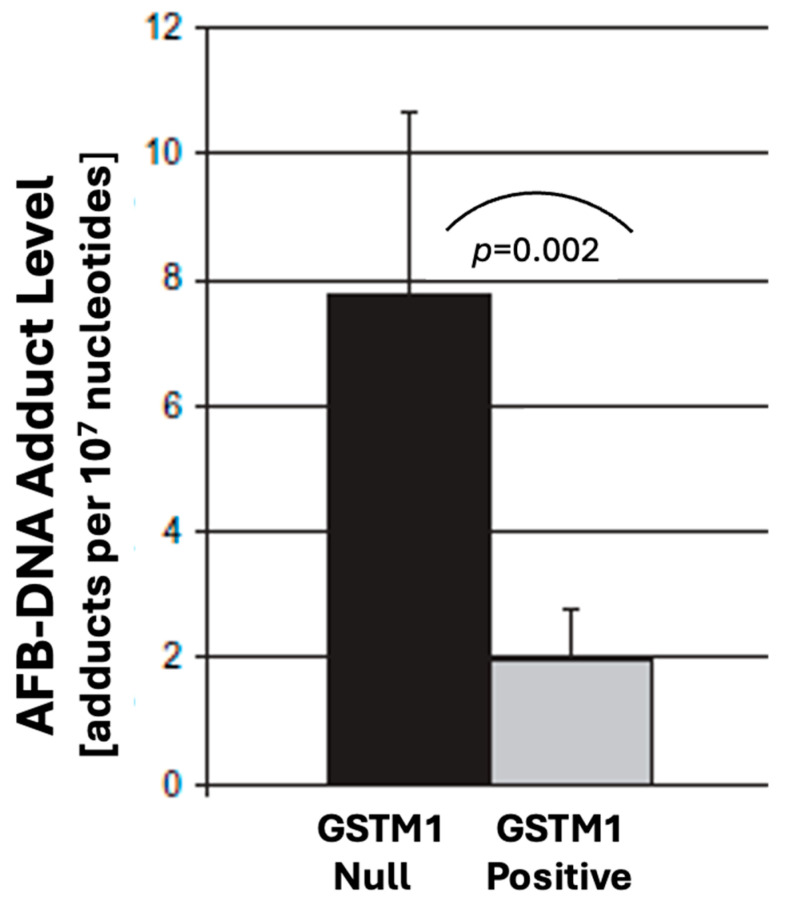
Modulation of AFB-DNA adduct formation in the context of the GSTM1 genotype status. A total of 11 different hepatocyte preparations were examined for AFB-DNA binding. Six of the samples were GSTM1-null and five were GSTM1-positive. AFB-DNA adducts per 10^7^ nucleotides were calculated and are shown. Each bar represents the mean and SEM. Statistical significance was determined by unpaired *t*-test with equal variances. Adapted from: Gross-Steinmeyer et al. [157]. Reprinted with permission from Oxford Press, Oxford, UK OX2 6DP; license #5923750542730, 7 December 2024.

**Figure 8 toxins-17-00030-f008:**
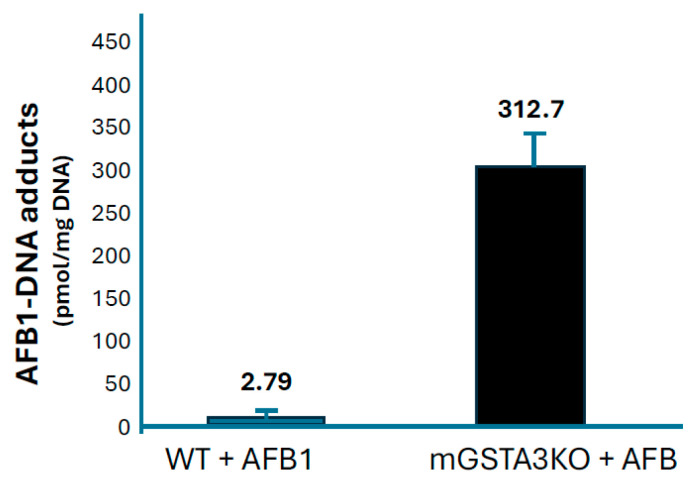
AFB-DNA adduct formation in mGstA3 knockout mice and wild-type. Mice (5 mGstA3 KO and 5 WT, 6 months of age, all males) were injected with a single dose of 5 mg/kg AFB1, dissolved in DMSO, in a volume of 100 μL/30 g of mouse weight, and euthanized 3 h later. Redrawn from: Ilic et al. [173], with permission from Elsevier Press, Berkeley, CA; license # 5923751463839, 7 December 2024.

**Figure 9 toxins-17-00030-f009:**
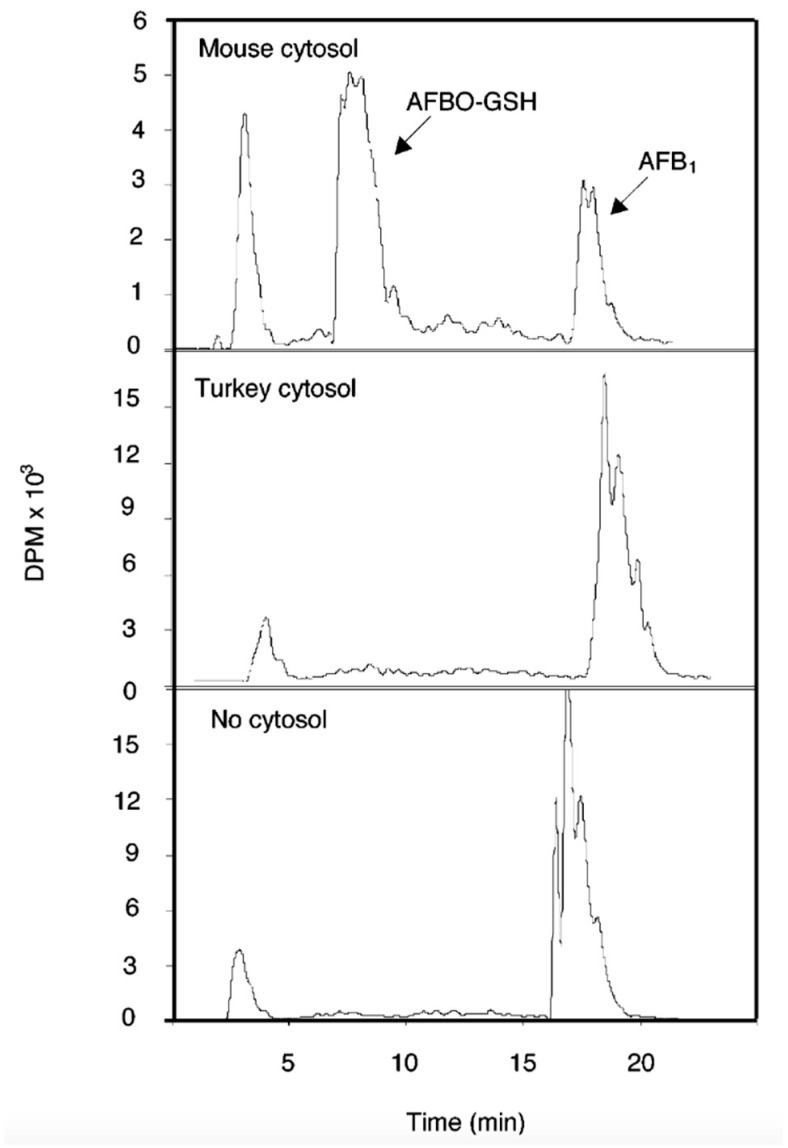
Reverse-phase HPLC radiochromatograms of cytosolic GST conjugation of AFBO in mouse and turkey. The top panel shows [^3^H]-AFBO-GST activity of BHA-induced mouse liver cytosol (500 mg protein) for comparison. The middle panel show the lack of GST-mediated [^3^H]-AFBO-conjugating ability of turkey hepatic cytosol (1200 mg protein). A control incubation with no cytosol is also presented (bottom panel). Even when a wide range of turkey cytosolic protein concentrations (400–1200 mg) was used, no GST-mediated trapping was detected [78]. Reprinted with permission from Elsevier Press, Berkeley, CA 94704, license # 5923770700438, 7 December 2024.

**Figure 10 toxins-17-00030-f010:**
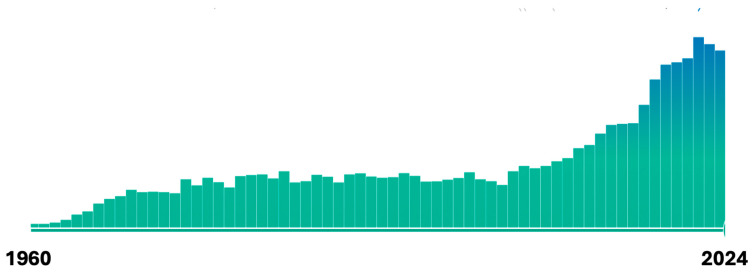
Timeline of research interest in aflatoxins, as indicated by the number of scientific publications each year from 1963 to December 2024. Data from a PubMed search on the term “aflatoxin” or “aflatoxins”.

**Figure 11 toxins-17-00030-f011:**
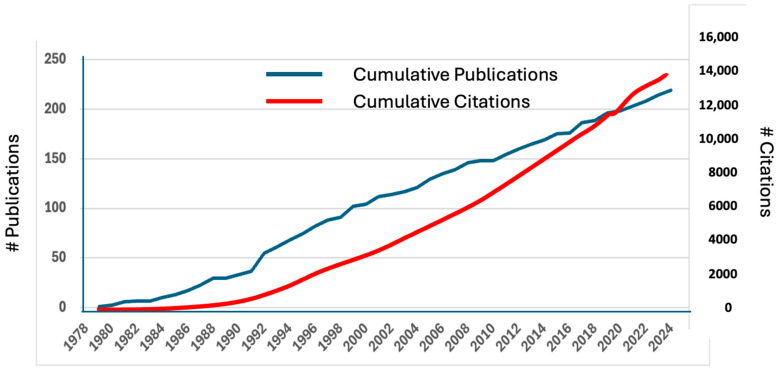
Publication and citation history of Dr. John Groopman’s contributions to the past 45 years of aflatoxin research, including many papers related to species differences in biotransformation. (Figure developed from data obtained from a Web of Science citation search on “John D. Groopman” and “aflatoxins”).

**Table 1 toxins-17-00030-t001:** Distribution of oxidative metabolites of AFB1 (1500 pmol/10^8^ cells, or approximately 20 µM) following 1 h of incubation in isolated rat hepatocytes. (Data from Ch’ih et al [50]).

AFB1 Form	% of Total	% Extracellular
Unchanged AFB1	5%	6%
Bound to DNA + proteins	23%	BDL
AFB-GSH	60%	72.4%
Distribution of Oxidative Metabolites		
AFBO (Bound + AFB-GSH)	83%	-
AFM1 (15% conjugated)	3%	92%
AFP1 (93% conjugated)	13%	95%
AFQ1 (<31% conjugated)	1%	89%

BDL: Below limit of detection.

**Table 2 toxins-17-00030-t002:** Metabolic profile [^14^C]-AFB1 and DNA binding of [^3^H]-AFB1 catalyzed by reconstituted rat CYP3A1 or CYP1A1 and trout CYP2K1 or CYP1A1.

Enzyme ^a^	Water Fraction ^b^	Peak 1 ^c^	AFBO	AFQ1	AFM1	TOTAL	DNA-Adducts ^d^
rCYP3A1	17	62	30	37	27	173	6
rCYP1A1	17	26	20	34	125	222	1
trCYP2K1	34	57	404	39	27	561	131
trCYP1A1	8	47	5	14	12	86	ND

^a^ CYPs (0.1 nmol) were reconstituted with rat NADPH P-450 oxidoreductase (0.2 nmol), DLPC (15 µg), SDS (50 µg), and NADPH (0.5 mM) in 50 mM Tris-HCL (pH 8.0) buffer containing EDTA (0.1 mM) and MgCl_2_ (15 µM) and incubated for 30 min at either 30° or 37 °C. Metabolites were quantified by scintillation counting of fractions co-eluting with standards. Values are in pmol/min/nmol CYP and represent the average of duplicates. ^b^ Water fraction represents [^14^C]-AFB1 radioactivity in the soluble fraction containing unidentified AFB1 metabolite(s). ^c^ Peak 1 is also an unknown early eluting peak containing unidentified AFB1 metabolite(s). ^d^ DNA adduction was determined using the same reconstitution system but with the addition of 150 µg calf thymus DNA and [^3^H]-AFB1 as substrate. Data are from Williams and Buhler [58].

**Table 3 toxins-17-00030-t003:** Species comparison of hepatic GST-mediated conjugation of microsomally generated and synthetic AFBO (From Slone et al. [152]. Reprinted with permission from Wolter Kluwers Health, Hagerstown MD, 21741; License #5924970823669, 9 December 2024).

Synthetic AFBO ^a^			Microsomally-Generated AFBO ^b^	
Species	pmol mg^−1^	Normalized to Rat	pmol min^−1^	Normalized to Rat
Human ^c^	1.8	0.02	<2 ^d^	<0.01 ^d^
Rat	88.4	1	140	1
Hamster	103	1.2	930	6.6
Mouse	571	6.5	7080	51

^a^ The final concentration of synthetic AFBO was 0.4 mM; ^b^ 125 µM AFB1 was used in the mouse microsomal AFBO generating System; ^c^ Equal volumes of 14 human liver cytosols were pooled for this assay; ^d^ Below the practical detection limit of assay (2 pmol min^−1^ per mg protein).

**Table 4 toxins-17-00030-t004:** Site-directed mutagenesis of rGSTA3 using amino acid substitutions to resemble conserved sites between high AFBO-activity enzymes mGstA3-3 and rGSTA5-5 ^a^.

Sequence Variant	AFBO-GST Activity	CDNB Activity
(Mutations of fGSTA3 to mGstA3)	nmol/mg/min	mmol/µg/min
mGstA3-3	265.0 ± 11.11 (34)	10.0 ± 0.26 (3)
rGSTA3-3	<0.02 ± 0.0 (9)	18.4 ± 1.04 (9)
rGSTA5-5	57.0 ± 2.32 (12)	9.8 ± 0.79 (6)
hGSTA1-1	<0.02 ± 0.0 (3)	57.1 ± 4.60 (6)
E208D	0.09 ± 0.10 (3)	5.0 ± 0.80 (3)
E208D + H108Y	2.1 ± 0.27 (8)	11.3 ± 1.19 (6)
E208D + H108Y + L207F	5.9 ± 0.56 (12)	48.0 ± 0.90 (9)
E208D + H108Y + L207F + E104I	8.4 ± 0.98 (10)	14.2 ± 0.40 (6)
E208D + H108Y + L207F + E104I + V217K	14 ± 0.62 (23)	23.6 ± 0.64 (6)
E208D + H108Y + L207F + E104I + V217K + Y111H	40 ± 3.09 (9)	8.7 ± 0.31 (9)

^a^ Data from Van Ness et al. [166].

**Table 5 toxins-17-00030-t005:** Specific GST activity and GST-mediated detoxification of AFBO by unique recombinant alpha-class GSTs and of hepatic cytosolic from domesticated and wild turkeys. Activities obtained from Swiss Webster mouse cytosol are provided for comparison.

Specific Enzyme Activity (nmol/min/mg Protein)					
	CDNB ^1^	DCNB ^2^	ECA ^3^	CHP ^4^	AFB1-GSH ^5^
tGST1.1 *	1674.61 ± 48.15 ^a^	16.35 ± 0.80 ^a^	44.31 ± 2.00 ^ab^	451.83 ± 12.41 ^ab^	19.54 ± 1.42 ^abc^
EWtGSTA1.1	850.45 ± 13.08b ^bc^	16.02 ± 0.58 ^a^	20.83±0.48 ^b^	1977.13 ± 11.45 ^de^	26.33 ± 0.89 ^de^
rGSTA1.2	7220.94 ± 75.81 ^b^	10.85 ± 0.09 ^b^	164.51 ± 6.18 ^c^	1076.7 ± 11.45 ^de^	22.49 ± 1.97 ^be^
EWtGSTA1.2	2761 ± 127.16 ^e^	5.70 ± 0.16 ^c^	48.80 ± 1.37 ^ad^	1423 ± 864.02 ^ef^	16.59 ± 2.10 ^acf^
RGWtGSTA1.2	3137.45 ± 234.93 ^f^	8.15 ± 0.61 ^d^	69.92 ± 0.96 ^d^	2002.85 ± 76.44 ^c^	16.62 ± 2.02 ^acf^
RPtGSTA1.2	3161.92 ± 15.18 ^f^	10.60 ± 0.43 ^b^	95.40 ± 7.36 ^e^	2769.28 ± 150.69 ^g^	21.97 ± 2.54 ^be^
tGSTA1.3	1063.48 ± 25.39 ^b^	1.46 ± 0.15 ^e^	50.58 ± 0.24 ^ad^	166.97 ± 68.47 ^a^	21.98 ± 2.26 ^be^
RGWtGSTA1.3	638.89 ± 12.60 ^cg^	1.40 ± 0.05 ^e^	23.61 ± 0.51 ^b^	1924.87 ± 486.10 ^cf^	29.10 ± 1.50 ^d^
tGSTA2	3045.23 ± 16.92 ^f^	7.55 ± 0.59 ^d^	34.49 ± 0.00 ^ab^	314.06 ± 7.29 ^a^	21.71 ± 0.54 ^be^
RGWtGSTA2	1880.01 ± 61.97 ^a^	9.31 ± 0.31 ^f^	29.58 ± 0.69 ^ab^	287.88 ± 7.91 ^a^	21.35 ± 0.50 ^ab^
RPtGSTA2	1709.40 ± 71.87 ^a^	7.23 ± 0.24 ^d^	35.85 ± 1.48 ^ab^	307.90 ± 8.29 ^a^	19.42 ± 0.94 ^abc^
tGSTA3	4254.39 ± 37.32 ^h^	1.25 ± 0.00 ^e^	174.32 ± 1.54 ^c^	850.25 ± 59.46 ^bd^	18.04 ± 2.67 ^abf^
RPtGSTA3	1649.30 ± 53.04 ^a^	1.23 ± 0.21 ^e^	543.28 ± 24.36 ^f^	1934.90 ± 30.79 ^cf^	21.42 ± 1.43 ^b^
tGSTA4	302.57 ± 44.57 ^ij^	0.71 ± 0.28 ^e^	21.38 ± 0.98 ^b^	27.66 ± 0.45 ^a^	20.06 ± 0.33 ^abc^
EWtGSTA4	504.92 ± 91.97 ^gf^	0.86 ± 0.27 ^e^	453.39 ± 15.09 ^g^	90.99 ± 3.49 ^a^	14.57 ± 1.56 ^f^
RGWtGSTA4	251.09 ± 8.98 ^j^	1.06 ± 0.42 ^e^	365.68 ± 14.88 ^h^	91.39 ± 3.52 ^a^	16.10 ± 1.29 ^cf^
Hepatic GSTs					
DT	1027.20 ± 17.81 ^a^	1.51 ± 0.05 ^a^	89.58 ± 0.80 ^a^	180.87 ± 0.97 ^a^	n.d.
EW	714.09 ± 37.10 ^b^	2.21 ± 0.30 ^a^	90.52 ± 3.75 ^a^	209.27 ± 13.56 ^a^	0.018 ± 0.004 ^a^
RGW	524.14 ± 14.53 ^c^	1.59 ± 0.15 ^a^	81.06 ± 1.98 ^a^	159.56 ± 11.03 ^a^	0.028 ± 0.007 ^a^
RP	890.18 ± 43.81 ^a^	2.61 ± 0.09 ^a^	94.78 ± 4.93 ^a^	203.91 ± 8.50 ^a^	0.017 ± 0.003 ^a^
Mouse	2888.69 ± 43.98 ^d^	64.55 ± 1.10 ^b^	61.10 ± 0.14 ^b^	805 ± 43.68 ^b^	0.740 ± 0.013 ^b^

Under each column, different superscript letters represent significant difference (*p* < 0.05). (A) Recombinant GSTs were analyzed by ANOVA and post hoc LSD was used for mean separation. 1. 1-Chloro-2,4-dinitrobenzene: GSH (1 mM); 2. 1,2-Dichloro-4-nitrobenzene: GSH (5 mM); 3. Ethacrynic acid; GSH (2.5 mM); 4. Cumene Hydroperoxide; GSH (2 mM); 5. AFB1-8,9-epoxide; GSH (5 mM); Mean 6 SD of triplicate determination. * Abbeviations: tGST: domestic turkey; EWtGST: Eastern Wild turkey; RGWtGST: Rio Grande Wild turkey; RPtGST: Royal Palm turkey. From: Kim et al. [178]. Reprinted under Open Access Creative Commons License.

## Data Availability

No new data were created or analyzed in this study.

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
