# Peer review of "Species Differences in the Biotransformation of Aflatoxin B1: Primary Determinants of Relative Carcinogenic Potency in Different Animal Species"

_toxins, 2025, doi:10.3390/toxins17010030_

Round 1

Reviewer 1 Report

Comments and Suggestions for Authors

Reviewing comments:

1. Redundancy and Length:

Combine redundant chapters and paragraphs for better flow and clarity.

Suggestions provided for specific sections to combine or move.

2.      Structure Improvements:

Revise the summary section to merge with the conclusion and remove unrelated new information.

3.      Content Presentation:

Add summary tables for each pathway (#1 to #8) to simplify and avoid repetitive text.

Ensure proper correlation between references and statements.

4.      Terminology and Formatting:

Use consistent formatting for terms like "AFB1," enzyme names ("CYP"), and other proteins.

Address spelling and formatting mistakes throughout.

5.      Additional Information:

Include information about biotransformation and toxicity in the intestines, specifically involving CYP enzymes.

6.      Title and Abstract:

Rephrase the title for clarity, avoiding confusion with fungal species.

Improve the abstract for professionalism and clarity

Major comments

1.      Combine Chapters:

·         Review chapters mentioned (e.g., 1237-1244) and combine overlapping content.

·         Move paragraphs (e.g., Line 927-928) to more suitable chapters for logical flow.

2.      Refine Summary and Conclusion:

·         Merge the summary and conclusion sections, ensuring they don’t introduce new information.

·         Revise lines 1392-1397 to avoid redundancy.

3.      Summary Tables:

·         Create summary tables under each pathway, clearly highlighting species differences.

4.      References and Correlation:

·         Verify the accuracy and relevance of cited references (e.g., Lines 119, 151, 435, 926, 1009).

·         Update older references with recent, high-impact studies.

5.      Intestinal Biotransformation:

·         Include details on biotransformation and toxicity in the intestines, with emphasis on CYP enzymes' roles.

Minor comments

1.         Citations:

·           Add proper citations for statements (e.g., Lines 36, 167, 452).

2.         Title and Abstract:

·           Rephrase the title to reflect "host species" for clarity.

·           Improve abstract language (e.g., Lines 16-18) using concise and professional terminology.

3.         Formatting and Terminology:

·           Standardize the use of terms (e.g., "AFB1," "CYP enzymes") across the manuscript.

·           Perform a thorough check for spelling and formatting issues.

Reviewer 2 Report

Comments and Suggestions for Authors

The manuscript provides a comprehensive review on the species differences in the biotransformation of AFB1, a significant mycotoxin with hepatotoxic and carcinogenic properties. The authors have extensively covered the literature on the topic, highlighting the role of various biotransformation pathways in determining the sensitivity of different species to AFB1. The manuscript will contribute to a better understanding of the complex issue of species differences in AFB1 metabolism and their implications for risk assessment and management. I recommend this manuscript for publication after the authors address the following comments and suggestions.

Line 57 and134: “aflatoxin B1”, the full name “aflatoxin B1” should be given in abbreviation.

Tables 3 and 5 were in the format of graphs, and they should be presented in the form of tables.

Line 349, delete the symbol “}”, Line 352, add the symbol “)” after the word “ducks”; lines 353, delete a symbol “[“.

Line 111, “Gerdemann et al., 2023 [6]”; Line 337, Buhler, 1983 [54]; line 433, Rawl and Coulombe 2011 [83]; Line 766, Kelly et al., 2002; [136]; Line 957, Gross-Steinmeyer et al. 2010[154]; Line 1095, Kim et al., 2013 [175]; the citation format of these references in the text is not standardized, and the year should be removed.

Line 172, “Kamden” should be “Kamdem”.

Line 180, “Gallagher et al., 2006” and Line 194 “Kamdem et al. (2006)”, the citations of these references in the text should be given reference numbers and delete the year

Line 435, “In 1967 Allcroft and Carnaghan [84,85]”, Reference 85 was published in 1963, so it is incorrect to state 1967 here.

Line 748, Guengerich et al. [32,134,135], The author of reference 135 is not Guengerich. Please delete it herein and supplement the citation of reference 135 in the main text.

Reference 98,126,129,152, in the title of the paper, the first letters of all words are capitalized and should be lowercase

Reference 160, incomplete information, please provide the names of the authors.

Reviewer 3 Report

Comments and Suggestions for Authors

Dear authors,

I really appreciate your extensive work, but in my opinion  this paper is more suitable as a book chapter than a review. If the authors still want to publish their paper as review, an extensive correction must be taken into account. I suggest two different ways to follow: (i) the review could be structured as a history of the data regarding the biotransformation of AFB1 or (ii) only the recent studies (since the year 2000) could be taken into account.

Author Response

We understand how this might be viewed as a 'book chapter' rather than a 'review'.  However, it is part of a series of articles, and will 'set the stage' for other articles that talk about Biomarkers, etc.  But to address this concern, we have changed the introductory materials to present this review in an historical context.  The comprehensive, but focused, review would be lost if we only focused on papers since the year 2000.  We think that seeing this paper as part of a series of papers in this Special Issue, as seen from an historical context, addresses the reviewer's concern.

Round 2

Reviewer 3 Report

Comments and Suggestions for Authors

I understand that this is a part of a series of articles.

Congrats to the authors for their comprehensive work! 

My final recommendation is: accept in this present form.